# No Task Left Behind:
# Isotropic Model Merging with Common and Task-Specific Subspaces

**Daniel Marczak** [1 2]  **Simone Magistri** [3]  **Sebastian Cygert** [4 5]
**Bartłomiej Twardowski** [6 7 8]  **Andrew D. Bagdanov** [3]  **Joost van de Weijer** [7 8]

## Abstract

Model merging integrates the weights of multiple task-specific models into a single multi-task model. Despite recent interest in the problem, a significant performance gap between the combined and single-task models remains. In this paper, we investigate the key characteristics of task matrices – weight update matrices applied to a pre-trained model – that enable effective merging. We show that alignment between singular components of task-specific and merged matrices strongly correlates with performance improvement over the pre-trained model. Based on this, we propose an isotropic merging framework that flattens the singular value spectrum of task matrices, enhances alignment, and reduces the performance gap. Additionally, we incorporate both common and task-specific subspaces to further improve alignment and performance. Our proposed approach achieves state-of-the-art performance on vision and language tasks across various sets of tasks and model scales. This work advances the understanding of model merging dynamics, offering an effective methodology to merge models without requiring additional training.

## 1. Introduction

Pre-trained models are the foundation of modern machine learning systems (Carion et al., 2020; Radford et al.,

[1]Warsaw University of Technology, Poland [2]IDEAS NCBR, Warsaw, Poland [3]Department of Information Engineering, Media Integration and Communication Center (MICC), University of Florence, Italy [4]NASK - PIB, National Research Institute, Warsaw, Poland [5]Gdańsk University of Technology, Poland [6]IDEAS Research Center, Warsaw, Poland [7]Computer Vision Center, Barcelona, Spain [8]Department of Computer Science, Universitat Autonoma de Barcelona, Spain. Correspondence to: Daniel Marczak <daniel.marczak.dokt@pw.edu.pl>.

*Proceedings of the 42nd International Conference on Machine Learning*, Vancouver, Canada. PMLR 267, 2025. Copyright 2025 by the author(s).

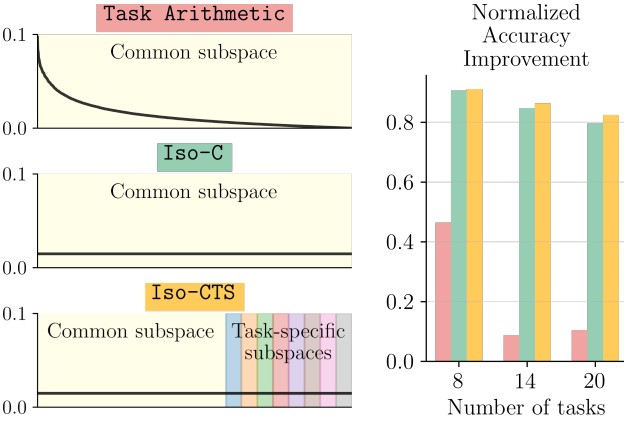

*Figure 1.* Spectrum of singular values for a single layer weight update matrix obtained by merging using `Task Arithmetic` (top) compared to our approaches: `Iso-C` (middle) and `Iso-CTS` (bottom). `Task Arithmetic` sums the task-specific matrices, which result in a spectrum with a few dominant components. `Iso-C` instead replaces this spectrum with a uniform one, which results in significant performance improvement. `Iso-CTS` enhances the common subspace with task-specific subspaces and yields state-of-the-art model merging performance.

2021; Caron et al., 2021; Zhai et al., 2023). In practice, they are typically fine-tuned for specialization on specific tasks (Wortsman et al., 2022b; Ilharco et al., 2022). Recently, a growing body of research has focused on *model merging* (Li et al., 2023), which combines multiple task-specific experts into a single multi-task model. Many methods have been proposed to improve the effectiveness of model merging by reducing sign conflicts (Yadav et al., 2023), by aligning gradients (Daheim et al., 2024), or through magnitude-based selection (Marczak et al., 2024). However, a significant performance gap between the combined and single-task models remains.

A key insight from Ilharco et al. (2023) is that *task vectors*, defined as the offset between the *flattened* fine-tuned weights and the pre-trained checkpoint, from different tasks are typically close to orthogonal. This orthogonality has been seen as a fundamental property enabling effective merging with reduced interference and has inspired works that en-

force the orthogonality by modifying the fine-tuning procedure (Po et al., 2024). Most recently, Stoica et al. (2025) and Gargiulo et al. (2025) have shown that accounting for the structure of the weight update matrix, dubbed *task matrix*, is a more effective strategy for improving the performance of model merging. In this paper, we investigate precisely what the characteristics of task matrices are that favor effective model merging. Different from previous works, we propose to analyze the alignment between task-specific and merged subspaces.

Specifically, to capture the similarity between task matrices, we propose to investigate the *Subspace Alignment Ratio*. Through the lens of Singular Value Decomposition, our metric quantifies the similarity between subspaces spanned by the top singular vectors of task matrices. When applied to compare matrices of the merged model to the task-specific ones, this metric strongly correlates with the performance of the merged model on a given task. This allows us to identify the directions amplified by multiple tasks as well as the underrepresented directions that lead to poor performance on corresponding tasks.

Our goal is to design a model merging technique that balances directions in the weight space across different tasks. We achieve this by flattening the singular values spectrum of the merged matrix, making it more uniform. Enforcing a uniform (isotropic) spectrum significantly improves the alignment and performance of the merged model. This simple yet effective adjustment, which requires no changes to the fine-tuning procedure, leads to substantial gains in merging performance (see method `Iso-C` in Figure 1).

However, tasks with dominant directions of smaller intensity compared to the majority of tasks and whose directions are orthogonal to the common directions may still remain underrepresented, especially when the number of tasks increases. To address this, we enhance isotropic model merging by introducing task-specific subspaces that retain unique task features while preserving shared knowledge. Our approach begins with the top singular values of the common subspace and iteratively replaces the least significant singular vectors with task-specific directions. This strategy allows us to increase the scalability of our merging approach to more tasks (see method `Iso-CTS` in Figure 1).

The main contributions of this paper are:

- We show that the alignment between the subspace spanned by the principal directions of the task-specific matrices and that of the merged matrix positively correlates with the performance of the merged model.

- We demonstrate that applying an isotropic scaling to singular directions of merged task matrices improves the alignment between merged and task-specific matrices. This results in a simple yet highly effective tech-

nique for model merging that we call `Iso-C`, which outperforms most baselines.

- We further enhance our approach by incorporating task-specific directions into the merged matrix resulting in `Iso-CTS`, a merging method that achieves state-of-the-art results, in particular for a large number of tasks.

- Our methods demonstrate versatility, achieving state-of-the-art on vision and language merging benchmarks for both fully and LoRA fine-tuned models[1].

## 2. Related Work

**Model merging.** Pre-trained models serve as a foundation for expert models specialized in specific downstream tasks (Radford et al., 2021). Recently, model merging has emerged as a promising technique to combine multiple expert models into a single multi-task model. One of the pioneering works in the field, Task Arithmetic (TA) (Ilharco et al., 2023), proposed to compute a *task vector* as a difference between the expert and the pre-trained model and to then aggregate task vectors via scaled addition to create an expert in multiple tasks. The significant performance gap between individual experts and the combined model sparked an abundance of works with the aim of reducing interference when merging models. TIES (Yadav et al., 2023) proposed a novel way to reduce sign conflicts between the parameters of expert models, Model Breadcrumbs (Davari & Belilovsky, 2024) removed outliers from the task vectors, and Consensus Merging (Wang et al., 2024b) removed catastrophic and selfish weights. These methods focused on per-parameter techniques to mitigate the interference, treating each parameter independently.

The aforementioned *static merging* methods output a single set of multi-task weights which can be used as a drop-in replacement for the pre-trained model. However, a number of recent methods, dubbed *dynamic merging*, alter the inference procedure to improve the results. Twin-Merging (Lu et al., 2024) composes task-specific components at test-time and alters the inference algorithm requiring two forward passes. EMR-Merging (Huang et al., 2024) uses additional per-task parameter masks and rescalers to perform inference. In this paper, we consider static merging exclusively.

**Singular Value Decomposition of model weights.** While SVD of weight matrices has been primarily used for model compression (Denton et al., 2014; Kim et al., 2016), recently its effectiveness was also identified for fine-tuning of large models. LoRA (Hu et al., 2021) uses SVD to identify the similarities of weight updates between low-rank and full-rank fine-tuning. MiLORA (Wang et al., 2024a) identifies

---

[1]The code is available at `https://github.com/danielm1405/iso-merging`.

that the bottom singular components correspond to noisy or long-tail information, while the top singular vectors contain important knowledge. Therefore, they propose a fine-tuning approach that updates only the minor singular components of the weight matrix while keeping the top singular components frozen. SVFT ([Lingam et al., 2024](#)) computes outer products of its singular vectors and, during fine-tuning updates, only sparse coefficients of these combinations.

**SVD for model merging.** The structure imposed by SVD was used for model merging in KnOTS ([Stoica et al., 2025](#)), which proposes to concatenate the task-specific low-rank adaptation matrices (LoRA) and average the right-singular vectors before SVD reconstruction to obtain the merged weights. The most similar work to us is the parallel work Task Singular Vectors (TSV) ([Gargiulo et al., 2025](#)), which measures task interference based on the interaction of singular vectors from different tasks and uses it to increase merging effectiveness. We share the motivation to improve model merging through SVD decomposition. However, while they focus on the orthogonalization of task-specific subspaces to reduce interference, we show that making singular values uniform in a common subspace is a surprisingly powerful method. Further, we show how to combine shared and task-specific subspaces for improved performance.

# 3. Background and Motivation

In this Section, we first describe the general framework of model merging and provide the notation used throughout the rest of the paper. We then motivate our approach via an analysis of the correlation between task similarity and performance improvement of the merged model.

## 3.1. Model Merging

Model merging integrates multiple deep neural network models, each individually trained (i.e. fine-tuned) on distinct tasks starting from the same pre-trained model, into a single merged model. Let $\theta_0$ denote the weights of the pre-trained network, and $\theta_t$ denote the fine-tuned weights for task $t$, with $t = 1, \ldots, T$, where $T$ is the total number of tasks. We will use the notation $\theta_t^{(l)}$ to identify the weights of layer $l$ for task $t$ and $L$ to denote the total number of layers in a network. The objective of model merging is to find a merging function $f$, such that the model:

$$\theta_M^{(\ell)} = f(\theta_0^{(\ell)}, \{\theta_t^{(\ell)}\}_{t=1}^T), \quad \forall \ell = 1, \ldots, L \quad (1)$$

is able to perform all tasks on which the individual models $\theta_t$ are trained.

Building upon Task Arithmetic (TA), we define the layer-wise *task matrix* $\Delta_t^{(\ell)}$ as the difference between the weights

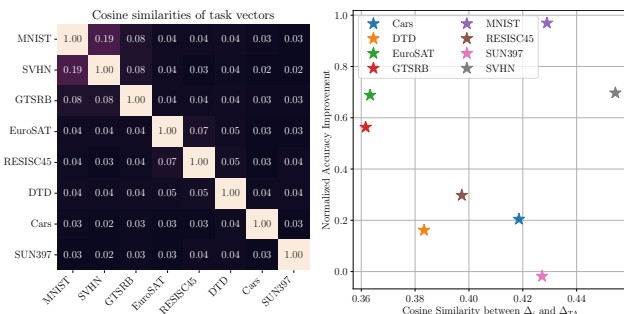

(a) Cosine similarity between pairs of task vectors.

(b) NAI vs cosine similarity between task and merged vectors.

*Figure 2.* (a) Tasks vectors are typically close to *orthogonal* to each other. (b) Models with very different normalized accuracy improvements (NAI) exhibit very close cosine similarities, and the correlation between cosine similarity and NAI is low.

of the model $\theta_t$ and the pre-trained model $\theta_0$ for layer $\ell$:

$$\Delta_t^{(\ell)} = \theta_t^{(\ell)} - \theta_0^{(\ell)}. \quad (2)$$

In the rest of the paper, the $\ell$ superscript is omitted when not relevant to the discussion, and all definitions refer to an arbitrary layer. The authors of Task Arithmetic propose to solve the problem of model merging by defining a merging function that sums all task matrices to the pre-trained model weights:

$$\theta_{TA}^{(\ell)} = \theta_0^{(\ell)} + \alpha \Delta_{TA}^{(\ell)}, \quad (3)$$

where $\alpha$ is a scaling factor determined on a held-out validation dataset and $\Delta_{TA}^{(\ell)} = \sum_{t=1}^T \Delta_t^{(\ell)}$. The advantage of this merging strategy is that it allows for the reuse and transfer of knowledge from many fine-tuned models to the pre-trained model without requiring additional training or access to the original training data ([Ilharco et al., 2023](#)).

## 3.2. Cosine Similarity and Performance Improvement are Uncorrelated

Starting from the definition of Task Arithmetic (TA) in Eq. ([3](#)), we aim to explore the possible reasons for the improvement achieved by TA merging over the pre-trained (or zero-shot) model across multiple tasks. To empirically quantify performance gain, we propose the *Normalized Accuracy Improvement (NAI)* metric, defined as:

$$\text{NAI}(\theta_M, \theta_t; \theta_0) = \frac{\text{Acc}(\theta_M) - \text{Acc}(\theta_0)}{\text{Acc}(\theta_t) - \text{Acc}(\theta_0)}, \quad (4)$$

which quantifies the improvement of the merged model $\theta_M$ relative to that achieved by the task-specific model $\theta_t$, both measured with respect to the zero-shot baseline $\theta_0$.[2]

---

[2]NAI differs from Normalized Accuracy ([Ortiz-Jiménez et al.,](#) [2023](#)) which does not account for zero-shot performance.

Ilharco et al. (2023) hypothesize that minimal inter-task interference – captured by near-zero cosine similarity between the vectorized representation of the task matrices, i.e., $\langle \text{vec}(\Delta_i), \text{vec}(\Delta_j) \rangle \approx 0$ for $i \neq j$ (see Figure 2a) – explains the effectiveness of Task Arithmetic. To investigate this further, we examine whether the cosine similarity between each task vector and the merged Task Arithmetic vector, $\langle \text{vec}(\Delta_{\text{TA}}), \text{vec}(\Delta_t) \rangle$, serves as an indicator of performance improvement, as quantified by $\text{NAI}(\theta_{\text{TA}}, \theta_t; \theta_0)$. However, we observe no clear correlation (see Figure 2b), suggesting that cosine similarity alone does not fully explain the observed performance gains. This indicates that the improvement achieved by Task Arithmetic likely originates from other factors, which we unveil below through spectral analysis of the Task Arithmetic and task-specific matrices.

### 3.3. Performance Correlates with Subspace Alignment

We argue that the improvement in Task Arithmetic performance derives from the relationship between the top singular vectors of $\Delta_{\text{TA}}$ and those of each $\Delta_t$. Specifically, we hypothesize that the subspace of $\Delta_{\text{TA}}$ approximates the union of the subspaces of each $\Delta_t$, and that the overlap of this overall subspace with each task matrix correlates with the performance improvement of the merged model.

In order to empirically quantify the overlap between subspaces, we propose the *Subspace Alignment Ratio (SAR)* metric. We define SAR between a task matrix $\Delta_t$ and a generic merged task matrix $\Delta_{\text{M}}$ as:

$$\text{SAR}(\Delta_t, \Delta_{\text{M}}; k_{\text{M}}) = \frac{||\Pi_{k_{\text{M}},\text{M}} \Delta_t||_F}{||\Delta_t||_F}, \tag{5}$$

where $\Pi_{k_{\text{M}},\text{M}} = U_{k_{\text{M}},\text{M}} U_{k_{\text{M}},\text{M}}^\top$ is the projection matrix onto the subspace spanned by the top $k_{\text{M}}$ left-singular vectors of $\Delta_{\text{M}}$. The columns of $U_{k_{\text{M}},\text{M}}$ are obtained from the SVD decomposition of $\Delta_{\text{M}}$, and the number of singular vectors used ($k_{\text{M}}$) is determined from the merged task matrix $\Delta_{\text{M}}$ by minimizing the approximation error with $\epsilon = 0.05$:

$$k_M = \min \left\{ k : ||\Delta_{\text{M}} - \Pi_{k,\text{M}} \Delta_{\text{M}}||_F \leq \epsilon \, ||\Delta_{\text{M}}||_F \right\}$$
$$= \min \left\{ k : \frac{\sum_{i=k+1}^r \sigma_i^2}{\sum_{i=1}^r \sigma_i^2} \leq \epsilon^2 \right\}, \tag{6}$$

where $\Sigma = \text{diag}(\sigma_1, \ldots, \sigma_r)$ contains the singular values of $\Delta_{\text{M}}$, and the equivalence follows from the definition of the Frobenius norm (see Appendix A.1).

SAR quantifies the alignment between the subspaces of two task matrices as a function of the number of dominant singular vectors of the merged matrix. To provide a single score measuring the overlap between two models, we denote with $\text{SAR}_{\text{avg}}$ the *Average Subspace Alignment Ratio* across all layers.

In Figure 3a (left, represented by stars), we plot the Normalized Accuracy Improvement achieved by TA on each task, given by $\text{NAI}(\theta_{\text{TA}}, \theta_t; \theta_0)$, against the Average Subspace Alignment Ratio of each task matrix $\Delta_t$ with the merged task matrix $\Delta_{\text{TA}}$, i.e. $\text{SAR}_{\text{avg}}(\Delta_t, \Delta_{\text{TA}}; k_{\text{TA}})$. First, we note that the alignment between task and merged matrices are notably high (ranging from 0.75 to 0.87), but vary significantly across datasets. This suggests that task vectors are well represented in the subspace identified by the task-arithmetic matrix but with different degrees of alignment and consistency depending on dataset characteristics. Furthermore, we highlight a strong correlation (Pearson correlation coefficient $\rho_{\text{TA}} = 0.94$) between the performance improvement on individual tasks achieved by $\theta_{\text{TA}}$ and the degree of alignment of $\Delta_t$ with $\Delta_{\text{TA}}$.

Analogous to the pairwise cosine similarity analysis between task vectors performed by Ilharco et al. (2023), in Figure 3b we measure the SAR between pairs of task matrices, $\text{SAR}_{\text{avg}}(\Delta_i, \Delta_j; k_{\text{TA}})$, using the $k_{\text{TA}}$ dominant components of the merged Task Arithmetic model. Some groups of tasks exhibit higher alignment which is due to their semantic similarity, e.g. MNIST, SVHN, and GTSRB are digit recognition datasets, while EuroSAT and RESISC45 are satellite image datasets. On the other hand, datasets such as Cars, DTD or SUN397 are less aligned to other tasks. Most importantly, tasks belonging to highly aligned groups are also highly aligned with the TA model and achieve the highest accuracy improvements (see Figure 3a). The tasks that are not aligned are underrepresented in the dominant subspace of $\Delta_{\text{TA}}$, and the performance on them is low.

Based on the observed correlation between performance and alignment ratio, we hypothesize that a merging method that aims to achieve high alignment will also achieve strong performance. Therefore, in the next Section, we propose an approach called *Isotropic Merging* that improves alignment and, most importantly, the performance of the merged models.

## 4. Isotropic Merging in Common and Task-specific Subspaces

In this Section, we propose a novel model merging method we call Isotropic Merging in Common and Task-Specific Subspaces (`Iso-CTS`). First, we introduce Isotropic Merging in Common Subspace (`Iso-C`), which is able to enhance the normalized accuracy improvement and the alignment of each task matrix using common directions identified by Task Arithmetic. Then, we show how to further enhance the performance of merged models by introducing task-specific directions to improve merging performance on sets of many diverse tasks.

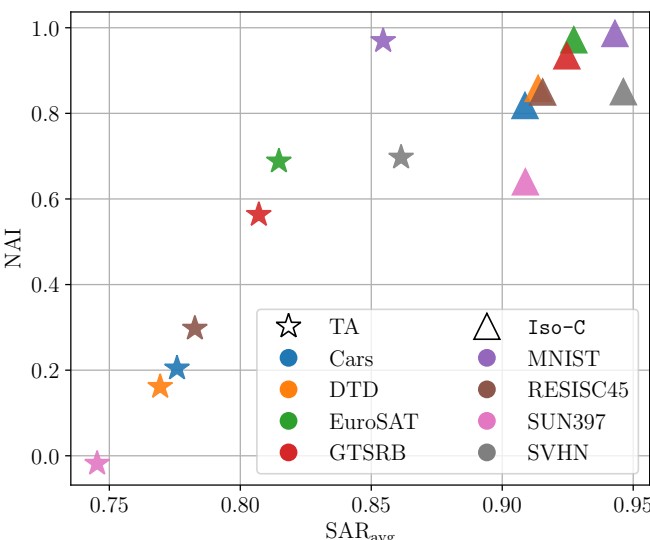

(a) Normalized Accuracy Improvement (NAI) vs. Average Subspace Alignment Ratio (SAR$_{avg}$).

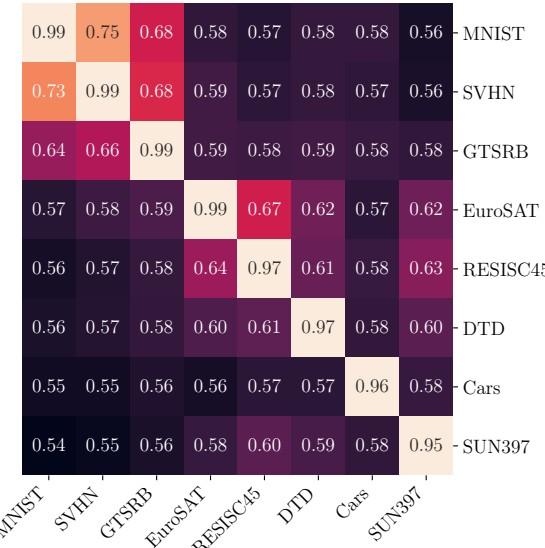

(b) Average Subspace Alignment Ratios (SAR$_{avg}$) between pairs of task matrices.

*Figure 3.* (a) NAI strongly correlates with SAR$_{avg}$ (Pearson correlation coefficient $\rho_{TA} = 0.94$). (b) Note the groups of highly aligned tasks such as {MNIST, SVHN, GTSRB} and {EuroSAT, RESISC45}. By comparing (b) and (a), the mutually aligned datasets exhibit higher alignment with the merged model and consequently achieve good performance. On the other hand, tasks with low mutual alignment, such as DTD, Cars, and SUN397, are less aligned with the merged model and achieve poor performance.

### 4.1. Isotropic Merging in Common Subspace

In Section 3.3, we demonstrated the high alignment of each task matrix with the matrix obtained by Task Arithmetic. This alignment indicates that the span of dominant singular vectors of the merged matrix effectively covers the subspace of each task and provides a good approximation of the *common subspace*. However, significant variability in the average alignment ratio across the dataset leads to a lower accuracy improvement for less aligned tasks compared to the tasks belonging to groups with high alignment. This variability stems from the skewness of the task arithmetic spectrum (Figure 1 and 12), which is concentrated in the first few singular values (which we call *top* or *dominant*), favoring the tasks from the highly aligned groups. Our proposed methodology, which we call *Isotropic Merging in Common Subspace* (Iso-C), aims to equalize the spectrum of the task arithmetic matrix in order to enhance the *average subspace alignment ratio* and ensure a more balanced representation across tasks in the merged model.

Consider the sum of task matrices $\Delta_{TA} = \sum_t \Delta_t$, where $\Delta_t \in \mathbb{R}^{m \times n}$. Via Singular Value Decomposition (SVD) on $\Delta_{TA}$ we obtain $\Delta_{TA} = U\Sigma V^\top$, where $U \in \mathbb{R}^{m \times r}$ and $V \in \mathbb{R}^{n \times r}$ represent, respectively, the left and right singular vectors of $\Delta_{TA}$, and $\Sigma \in \mathbb{R}^{r \times r}$ is the diagonal matrix containing the singular values. We denote the vector of singular values by $\sigma = \text{diag}(\Sigma) \in \mathbb{R}^r$.

---

**Algorithm 1** Iso-C: Isotropic Merging in Common Subspace

**Require:** Task matrices $\Delta_1, \ldots, \Delta_T$ with $\Delta_t \in \mathbb{R}^{m \times n}$
1: Sum task matrices: $\Delta_{TA} = \sum_{t=1}^T \Delta_t$
2: Compute the SVD of $\Delta_{TA}$: $\Delta_{TA} = U\Sigma V^\top$, with $U \in \mathbb{R}^{m \times r}, \Sigma \in \mathbb{R}^{r \times r}, V \in \mathbb{R}^{n \times r}, \sigma = \text{diag}(\Sigma) \in \mathbb{R}^r$
3: Calculate isotropic factor: $\overline{\sigma} = \frac{1}{r}\sum_{i=1}^r \sigma_i$ (Eq.7)
4: Reconstruct the matrix: $\Delta_{\text{Iso-C}} = \overline{\sigma}UV^\top$ (Eq.8)
5: **return** $\Delta_{\text{Iso-C}}$

---

To reduce the skewness towards the dominant singular vectors of $\Delta_{TA}$, we propose scaling all directions of the transformation applied by the right-singular vectors $V$ to a fixed value rather than using their corresponding singular values. This ensures that the final transformation is *isotropic*, with the scaling factor set to the average singular value:

$$\overline{\sigma} = \frac{1}{r}\sum_{i=1}^r \sigma_i, \quad (7)$$

and merged matrix is computed using the reconstruction:

$$\Delta_{\text{Iso-C}} = \overline{\sigma}UV^\top. \quad (8)$$

We apply this operation to all network layers, and the final merged model is defined as:

$$\theta_{\text{Iso-C}}^{(\ell)} = \theta_0^{(\ell)} + \alpha\Delta_{\text{Iso-C}}^{(\ell)}, \quad \forall\ell = 1, \ldots, L \quad (9)$$

**Algorithm 2** `Iso-CTS`: Isotropic Merging in Common and Task-Specific Subspaces (green – shared with `Iso-C`)

---

**Require:** Task matrices $\Delta_1, \ldots, \Delta_T$ with $\Delta_t \in \mathbb{R}^{m \times n}$

1: Sum task matrices $\Delta_{\text{TA}} = \sum_{t=1}^{T} \Delta_t$
2: Compute the SVD of $\Delta_{\text{TA}}$: $\Delta_{\text{TA}} = U\Sigma V^{\top}$, with $U \in \mathbb{R}^{m \times r}, \Sigma \in \mathbb{R}^{r \times r}, V \in \mathbb{R}^{n \times r}, \sigma = \text{diag}(\Sigma) \in \mathbb{R}^r$
3: **Retain** top-$k$ singular vectors and values from **common subspace**: $U^{1:k} = [u_1 | \ldots | u_k] \quad V^{1:k} = [v_1 | \ldots | v_k]$ $\sigma^{\text{cm}} = \text{diag}(\Sigma)^{1:k}$
4: **Accumulate** task-specific directions via projection:
5: **for** $t = 1$ to $T$ **do**
6: $\quad \overline{\Delta}_t = \Delta_t - U^{1:k}(U^{1:k})^{\top}\Delta_t$ $\qquad$ (Eq.10)
7: $\quad$ Compute SVD: $\overline{\Delta}_t = \overline{U}_t\overline{\Sigma}_t\overline{V}_t^{\top}$
8: $\quad$ Retain first $s = \frac{r-k}{T}$ components of $\overline{U}_t$ and $\overline{V}_t$: $\overline{U}_t^{1:s} = [\overline{u}_{t,1}|\ldots|\overline{u}_{t,s}] \quad \overline{V}_t^{1:s} = [\overline{v}_{t,1}|\ldots|\overline{v}_{t,s}]$ $\sigma_t^{\text{ts}} = \text{diag}(\overline{\Sigma}_t)^{1:s}$
9: **end for**
10: **Combine** common and task-specific spaces:
$$U_* = [U^{1:k}|\overline{U}_1^{1:s}|\ldots|\overline{U}_T^{1:s}] \in \mathbb{R}^{m \times r}$$
$$V_* = [V^{1:k}|\overline{V}_1^{1:s}|\ldots|\overline{V}_T^{1:s}] \in \mathbb{R}^{n \times r}$$
11: **Orthogonalize** $U_*$ and $V_*$ via whitening $\qquad$ (Eq.11)
12: Calculate isotropic factor $\overline{\sigma}$:
$$\overline{\sigma} = \frac{1}{r}\Big(\sum_{i=1}^{k}\sigma_i^{\text{cm}} + \sum_{t=1}^{T}\sum_{i=1}^{s}\sigma_{t,i}^{\text{ts}}\Big) \qquad \text{(Eq.13)}$$
13: **Reconstruct** the matrix $\Delta_{\text{Iso-CTS}} = \overline{\sigma}U_*V_*^{\top}$ $\quad$ (Eq.12)
14: **return** $\Delta_{\text{Iso-CTS}}$

---

where $\alpha$ is chosen on a held-out validation set.

Applying isotropic merging results in an enhancement of the normalized accuracy improvement and subspace alignment ratio (SAR) compared to Task Arithmetic (see Figure 3a). The increase in SAR is due to a higher number of dominant components $k_{\text{Iso-c}}$ in $\Delta_{\text{Iso-c}}$ (see Equation (6)), derived from the singular vectors of $\Delta_{\text{TA}}$, which are aligned with the subspaces of individual tasks (see Appendix A.2 for details). In Appendix A.3, we show that increased SAR is associated with reduced inter-task interference, measured by changes in internal activations induced by merging. In Algorithm 1, we present the `Iso-C` algorithm for a single layer.

### 4.2. Isotropic Merging in Common and Task-Specific Subspaces

The effectiveness of `Iso-C` depends on how well the common subspace – identified by the dominant singular vectors of $\Delta_{\text{TA}}$ – approximates the subspaces of the individual tasks. The approximation error arises from how these tasks interact when summed. The top singular directions of $\Delta_{\text{TA}}$ capture only the dominant common variations, while singular vectors associated with near-zero singular values provide

negligible information. At the same time, tasks with dominant directions of smaller intensity compared to the majority of tasks and whose directions are orthogonal to the common directions remain underrepresented. This limitation becomes more pronounced as the number of tasks increases and the tasks become more diverse (see Appendix A.4 for an extended discussion).

To address this limitation, we propose enhancing the range of directions used by `Iso-C` to ensure that the task-specific directions, which are orthogonal to those of the common subspace, are incorporated into the singular basis of the final merged matrix. We call this methodology as *Isotropic Merging in Common and Task-Specific Subspaces* (`Iso-CTS`).

Our approach starts with the top singular values of the common subspace and iteratively replaces the singular vectors associated with the lowest singular values with task-specific directions. The final goal is to find two orthonormal matrices $U_* \in \mathbb{R}^{m \times r}$ and $V_* \in \mathbb{R}^{n \times r}$ whose columns contain both common and task-specific directions. Afterward, the final matrix is reconstructed, and isotropic merging is applied. In the following, we provide a detailed explanation of our proposed algorithm.

**Retaining components from the common subspace.** We retain the top-$k$ singular vectors associated with the subspace identified by $\Delta_{\text{TA}}$:

$$U^{1:k} = [u_1 | \ldots | u_k] \quad V^{1:k} = [v_1 | \ldots | v_k],$$

where $U^{1:k}$, $V^{1:k}$ are the top-$k$ left- and right-singular vectors from the SVD of $\Delta_{\text{TA}}$. We analyze the impact of selecting $k$ in Section 5.4.

**Accumulating task-specific directions.** We project each task-specific matrix $\Delta_t$ onto the subspace orthogonal to the common subspace, i.e. the space spanned by top left-singular directions of the common subspace $U^{1:k}$:

$$\overline{\Delta}_t = \Delta_t - U^{1:k}(U^{1:k})^T\Delta_t. \qquad (10)$$

We then compute the SVD of $\overline{\Delta}_t = \overline{U}_t \overline{\Sigma}_t \overline{V}_t$ and retain the top $s = \frac{r-k}{T}$ directions for each task $t$:

$$\overline{U}_t^{1:s} = [\overline{u}_{t,1}|\ldots|\overline{u}_{t,s}] \quad \overline{V}_t^{1:s} = [\overline{v}_{t,1}|\ldots|\overline{v}_{t,s}], \forall t = 1, \ldots, T.$$

The orthogonal projection Eq. (10) guarantees that both the left- and right-singular vectors of $\overline{\Delta}_t$, representing task-specific directions, are orthogonal to the subspace spanned by the common directions (given by $U^{1:k}$).

**Combining common and task-specific matrices.** After identifying the $k$ principal vectors for the common subspace and $s = \frac{r-k}{T}$ principal vectors for each task, we now combine the common and task-specific directions by concatenating them: $U_* = [U^{1:k}|\overline{U}_1^{1:s}|\ldots|\overline{U}_T^{1:s}] \in \mathbb{R}^{m \times r}$ and $V_* = [V^{1:k}|\overline{V}_1^{1:s}|\ldots|\overline{V}_T^{1:s}] \in \mathbb{R}^{n \times r}$.

*Table 1.* Iso-CTS achieves state-of-the-art performance for all backbones on all evaluated scenarios. We present average absolute accuracy and average normalized accuracy (in subscript) in %. The best method in **bold** and the second-best underlined.

| Method | ViT-B/32 | | | ViT-B/16 | | | ViT-L/14 | | |
|---|---|---|---|---|---|---|---|---|---|
| | 8 tasks | 14 tasks | 20 tasks | 8 tasks | 14 tasks | 20 tasks | 8 tasks | 14 tasks | 20 tasks |
| Zero-shot | $48.3$ | $57.2$ | $56.1$ | $55.3$ | $61.3$ | $59.7$ | $64.7$ | $68.2$ | $65.2$ |
| Fine-tuned | $92.8$ | $90.9$ | $91.3$ | $94.6$ | $92.8$ | $93.2$ | $95.8$ | $94.3$ | $94.7$ |
| Weight Averaging | $66.3_{(72.1)}$ | $64.3_{(71.1)}$ | $61.0_{(67.5)}$ | $72.2_{(76.6)}$ | $69.5_{(74.8)}$ | $65.3_{(70.4)}$ | $79.6_{(83.2)}$ | $76.7_{(81.1)}$ | $71.6_{(75.6)}$ |
| Task Arithmetic | $70.8_{(76.5)}$ | $65.3_{(72.1)}$ | $60.5_{(66.8)}$ | $75.4_{(79.6)}$ | $70.5_{(75.9)}$ | $65.8_{(70.8)}$ | $84.9_{(88.7)}$ | $79.4_{(84.0)}$ | $74.0_{(78.1)}$ |
| TIES | $75.1_{(81.0)}$ | $68.0_{(74.8)}$ | $63.4_{(69.9)}$ | $79.7_{(84.3)}$ | $73.2_{(78.7)}$ | $68.2_{(73.3)}$ | $86.9_{(90.7)}$ | $79.5_{(84.1)}$ | $75.7_{(79.8)}$ |
| Consensus TA | $75.0_{(80.8)}$ | $70.4_{(77.4)}$ | $65.4_{(72.0)}$ | $79.4_{(83.9)}$ | $74.4_{(79.9)}$ | $69.8_{(74.9)}$ | $86.3_{(90.1)}$ | $82.2_{(86.9)}$ | $79.0_{(83.2)}$ |
| TSV-M | $85.9_{(92.3)}$ | $80.1_{(87.9)}$ | $77.1_{(84.3)}$ | $89.0_{(93.9)}$ | $84.6_{(91.0)}$ | $80.6_{(86.5)}$ | $93.0_{(97.0)}$ | $89.2_{(94.4)}$ | $87.7_{(92.5)}$ |
| **Iso-C (Ours)** | $\mathbf{86.3_{(92.9)}}$ | $80.3_{(88.1)}$ | $\underline{75.5_{(82.5)}}$ | $90.6_{(95.6)}$ | $84.8_{(91.1)}$ | $\underline{79.6_{(85.4)}}$ | $94.2_{(98.3)}$ | $89.3_{(94.5)}$ | $\underline{87.6_{(92.2)}}$ |
| **Iso-CTS (Ours)** | $\underline{86.2_{(92.8)}}$ | $\mathbf{81.7_{(89.7)}}$ | $\mathbf{78.1_{(85.5)}}$ | $\mathbf{91.1_{(96.1)}}$ | $\mathbf{86.4_{(92.8)}}$ | $\mathbf{82.4_{(88.4)}}$ | $\mathbf{94.7_{(98.8)}}$ | $\mathbf{91.0_{(96.3)}}$ | $\mathbf{90.1_{(94.9)}}$ |

**Orthogonalization.** There is no guarantee that the left- and right-singular task-specific vectors are orthogonal to each other, as we are only projecting each task matrix onto the common subspace. To reconstruct the final merged matrix, we must orthogonalize $U_*$ and $V_*$. Following Gargiulo et al. (2025), we compute the SVD of $U_* = P_{U_*} \Sigma_{U_*} Q_{U_*}^\top$ and $V_* = P_{V_*} \Sigma_{V_*} Q_{V_*}^\top$, and whiten (Schönemann, 1966):

$$U_* = P_{U_*} Q_{U_*}^\top \quad V_* = P_{V_*} Q_{V_*}^\top. \quad (11)$$

**Isotropic scaling and reconstruction.** Finally, we reconstruct the final merged matrix and apply isotropic merging:

$$\Delta_{\texttt{Iso-CTS}} = \overline{\sigma} U_* V_*^\top, \quad (12)$$

where $\overline{\sigma}$ is obtained by averaging the singular values associated with the vectors selected for both common and task-specific subspaces. Specifically, defining $\sigma^{\text{cm}} = \text{diag}(\Sigma)^{1:k} \in \mathbb{R}^k$, the vector of singular values associated with the common subspace identified by $U_{1:k}$ and $V_{1:k}$, and $\sigma_t^{\text{ts}} = \text{diag}(\overline{\Sigma}_t)^{1:s} \in \mathbb{R}^s$, with $s = \frac{r-k}{T}$, the vector of singular values associated with each task-specific subspace $\overline{U}_t^{1:s}$ and $\overline{V}_t^{1:s}$, we define the scaling factor as:

$$\overline{\sigma} = \frac{1}{r}\Big( \sum_{i=1}^{k} \sigma_i^{\text{cm}} + \sum_{t=1}^{T} \sum_{i=1}^{s} \sigma_{t,i}^{\text{ts}} \Big). \quad (13)$$

Finally, similar to ISO-C, the merged model is defined as:

$$\theta_{\texttt{Iso-CTS}}^{(\ell)} = \theta_0^{(\ell)} + \alpha \Delta_{\texttt{Iso-CTS}}^{(\ell)}, \quad \forall \ell = 1, \dots, L \quad (14)$$

where $\alpha$ is chosen on a held-out validation set.

## 5. Experimental Results

### 5.1. Fully fine-tuned vision models

We evaluate our approaches over sets of 8, 14, and 20 datasets, following Wang et al. (2024b). We provide the details of the datasets in Appendix C.1. We consider three

variants of CLIP (Radford et al., 2021) with ViT-B/32, ViT-B/16 and ViT-L/14 as visual encoders (Dosovitskiy et al., 2021). We use the checkpoints fine-tuned on the tasks above, provided in Wang et al. (2024b) (see Appendix D.1 for results using TA checkpoints). If not stated otherwise, we present the results using the ViT-B/16 visual encoder.

We compare our approaches with the following model merging methods: weight averaging (Wortsman et al., 2022a), Task Arithmetic (Ilharco et al., 2023), TIES-Merging (Yadav et al., 2023), Consensus TA (Wang et al., 2024b) and TSV-M (Gargiulo et al., 2025). We include the results of the zero-shot model and fine-tuned models serving as lower- and upper-bound, respectively. We compare the results based on absolute and normalized accuracy following standard practice (Wang et al., 2024b; Gargiulo et al., 2025).

Table 1 presents our main results for multi-task model merging. Iso-CTS achieves state-of-the-art results in all of the settings. Iso-C achieves very similar results to Iso-CTS in the 8 task scenario. However, Iso-CTS significantly outperforms Iso-C when merging 14 and 20 models, with improvements of up to 2.8% in absolute accuracy. This suggests that it is possible to faithfully represent a small number of tasks in the common subspace. However, when the number of tasks increases, it becomes crucial to retain important directions from the task-specific subspaces in order to maximize model merging effectiveness.

### 5.2. LoRA-adapted vision models

To evaluate our approaches in low-rank adaptation scenario, we follow the evaluation protocol of KnOTS (Stoica et al., 2025), a recent state-of-the-art method for merging LoRA fine-tuned models. We use codebase and checkpoints provided by KnOTS: ViT-B/32 and ViT-L/14 fine-tuned with rank 16 LoRA (Hu et al., 2021) on 8 vision tasks. To adapt our methodologies to low-rank regime, we simply operate on reconstructed task matrices, i.e. $\Delta W_t = B_t A_t$, where $A_t, B_t$ are LoRA matrices for task $t$. We compare Iso-C and Iso-CTS with TIES and DARE-TIES (Yu et al., 2024)

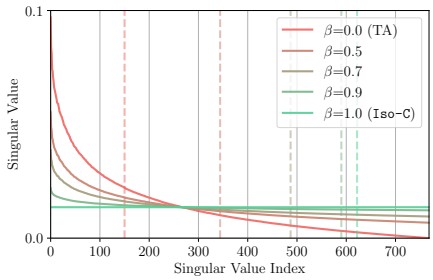

(a) Spectra of singular values for different values of interpolation coefficient ($\beta$).

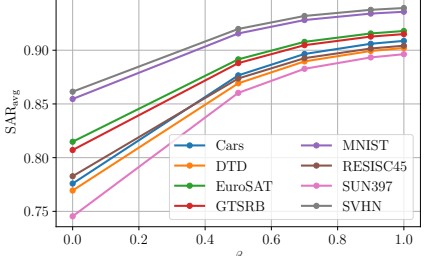

(b) Average Subspace Alignment Ratio (SAR$_{\text{avg}}$) vs. interpolation coefficient ($\beta$).

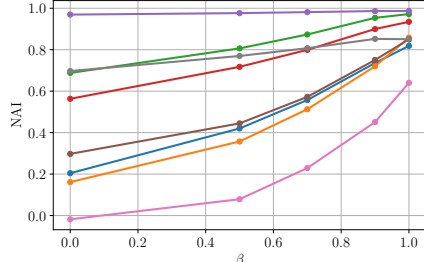

(c) Normalized Accuracy Improvement (NAI) vs. interpolation coefficient ($\beta$).

*Figure 4.* (a) Interpolating from $\Delta_{\text{TA}}$ ($\beta = 0$) towards $\Delta_{\text{Iso-C}}$ ($\beta = 1$) makes the spectrum of singular values of $\Delta_{\text{M}}$ more uniform and increases the number of preserved components $k_{\text{M}}$ (Eq. (6)) denoted by dashed lines. (b) This results in an increased alignment between each task-specific model and merged model measured by SAR$_{\text{avg}}$. (c) As alignment increases, the performance also improves as predicted based on the strong correlation between these two properties investigated in Section 3.3.

*Table 2.* Normalized per-task average accuracy. We merge 8 models fine-tuned with LoRA following (Stoica et al., 2025).

| Method | ViT-B/32 | ViT-L/14 |
|---|---|---|
| TA | 63.7 | 74.4 |
| TIES | 63.7 | 75.2 |
| DARE-TIES | 63.7 | 74.7 |
| KnOTS-TIES | 68.0 | 78.2 |
| KnOTS-DARE-TIES | 63.9 | 75.6 |
| **Iso-C (Ours)** | 73.6 | 83.7 |
| **Iso-CTS (Ours)** | **73.7** | **85.3** |

– combined with KnOTS or not – and TA.

We present the results in Table 2. Our methods, which are general purpose merging techniques, significantly outperform KnOTS, which are specifically designed for the LoRA merging. This highlights the versatility of Iso methods.

### 5.3. Language models

We present NLP results following the experimental setup from MaTS (Tam et al., 2023). We use T5-Large-LM-Adapt (Lester et al., 2021) base model (a variant of T5-Large (Raffel et al., 2020)) fine-tuned on subsets of 8 and 7 NLP tasks from T0 mixture (Sanh et al., 2022). We compare our approaches with weight averaging, TA, TIES, Fisher Merging (Matena & Raffel, 2021), RegMean (Jin et al., 2023), and MaTS (Tam et al., 2023).

We present the results in Table 3. Both Iso-C and Iso-CTS significantly outperform the competing approaches, which highlights the versatility of our proposed methods. We observe that Iso-CTS achieves very similar results to Iso-C suggesting that the common space captures all the directions necessary to reliably represent these 7 and 8 NLP tasks.

### 5.4. Analysis and Ablations

All the experiments in this Section are conducted on fully fine-tuned ViT-B/16 models. In Appendix B we provide the

*Table 3.* NLP results using T5-Large-LM-Adapt fine-tuned on tasks from T0 mixture. We present average absolute accuracy.

| Method | 8 tasks
(Zhou et al., 2022) | 7 tasks
(Yadav et al., 2023) |
|---|---|---|
| Fine-tuned | 80.7 | 85.9 |
| Weight Averaging | 56.4 | 60.5 |
| Task Arithmetic | 63.8 | 69.2 |
| TIES | 62.8 | 71.9 |
| Fisher Merging | 57.7 | 61.0 |
| RegMean | 69.1 | 74.3 |
| MaTS | 72.5 | 81.5 |
| **Iso-C (Ours)** | **75.6** | **83.3** |
| **Iso-CTS (Ours)** | 75.2 | 82.8 |

computational complexity analysis of our approaches.

**From Task Arithmetic to Isotropic Merging.** We analyze what happens when interpolating between the singular values obtained by Task Arithmetic (TA) and those obtained by Iso-C, i.e. the model with the following spectra:

$$\Sigma_\beta = (1 - \beta)\Sigma_{\text{TA}} + \beta\Sigma_{\text{Iso-C}}, \qquad (15)$$

where $\beta$ is an interpolation coefficient. Firstly, Figure 4a presents the change in singular values spectrum as we interpolate towards $\Delta_{\text{Iso-C}}$ ($\beta \rightarrow 1$). The skewed spectrum achieved by Task Arithmetic becomes isotropic, i.e. the scaling factor is equal along all of the singular directions. In Figure 4b we observe a steady increase in alignment between task-specific and merged models as measured by SAR$_{\text{avg}}$ (Eq. (5)), and Figure 4c shows that as alignment increases (with $\beta \rightarrow 1$), the performance of the merged model improves across all tasks. These results are consistent with our findings from Section 3.3 that show a strong correlation between alignment and the performance of the final model.

**The impact of singular directions on performance.** We analyze which singular directions contribute to the improvement of individual tasks. We truncate the flattened spectrum of Iso-C, keeping the $k$ directions associated with the leftmost singular values, i.e. $\sigma_i = \overline{\sigma}$ for $i \leq k$ and $\sigma_i = 0$

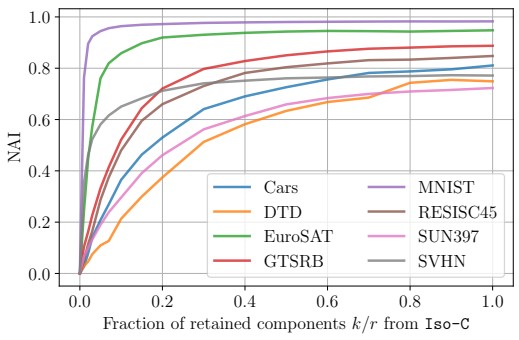

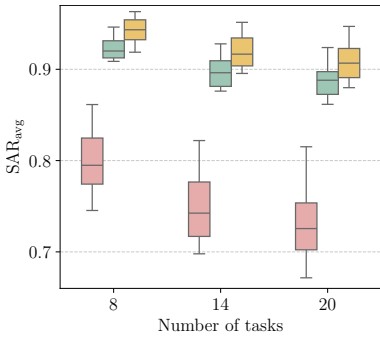

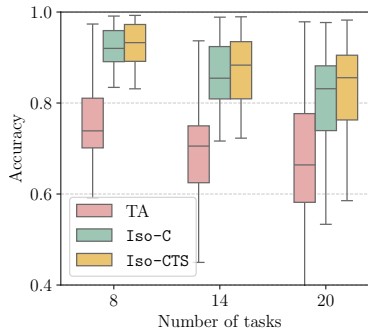

(a) Normalized Accuracy Improvement (NAI) of a model created by retaining $k$ components of Iso-C (associated with top-$k$ singular vectors from $\Delta_{\text{TA}}$).

(b) Average Subspace Alignment Ratios (SAR$_{\text{avg}}$) between merged and task-specific models for varying sets of tasks.

(c) Distribution of accuracies of the merged models for varying sets of tasks.

*Figure 5.* (a) The directions associated with the least significant singular values of $\Delta_{\text{TA}}$ have a minor contribution to the performance of Iso-C model. (b) Task-specific directions introduced in Iso-CTS improve the Average Subspace Alignment Ratio (SAR$_{\text{avg}}$) between task-specific models and the merged model compared to Iso-C which uses only a common subspace. (c) Higher alignment translates to higher accuracy of Iso-CTS with respect to Iso-C.

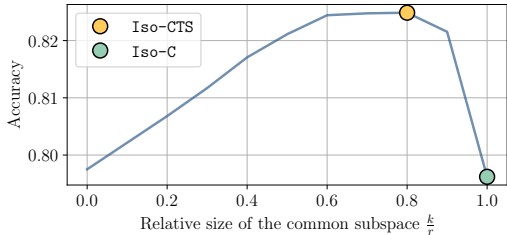

*Figure 6.* Iso-CTS is robust to the selected size of the common subspace as any value leads to improvement over Iso-C. These results are for the 20-task scenario.

for $i > k$. Note that the leftmost $k$ directions are the ones associated with the highest singular values of $\Delta_{\text{TA}}$. We plot the task-wise Normalized Accuracy Improvement (NAI, Eq. (4)) for varying $k$ in Figure 5a. We observe that the first few directions are responsible for rapid improvement on several tasks. Notably, these tasks belong to the aligned groups identified in Section 3.3 such as {MNIST, SVHN, GTSRB} and {EuroSAT, RESISC45}. Moreover, the directions associated with the least significant singular values of $\Delta_{\text{TA}}$ have a negligible contribution to the performance. This supports our intuition for replacing less significant common directions with task-specific components in Iso-CTS (see Section 4.2). Figure 5b shows that Iso-CTS achieves higher Average Subspace Alignment Ratio (SAR$_{\text{avg}}$, Eq. (5)) than Iso-C. Most importantly, Figure 5c shows that thanks to the addition of task-specific directions, Iso-CTS achieves better performance across tasks.

**Size of the common subspace for Iso-CTS.** While Iso-C operates only in the common subspace, Iso-CTS enhances it with task-specific subspaces. Therefore, we must select the size of the common subspace $k$ (and consequently the size of each task-specific subspace given by $\frac{r-k}{T}$). Figure 6 plots the relationship between accu-

racy and the fraction of subspace assigned for the common subspace ($\frac{k}{r}$) when merging 20 tasks. When $\frac{k}{r} = 1$ Iso-CTS is equivalent to Iso-C and suffers a 2.8% drop in accuracy from the maximum. The optimal fraction of common subspace $\frac{k}{r} = 0.8$, and we use this as a default value for Iso-CTS across all settings. Moreover, note that Iso-CTS is quite robust to the selection of this hyperparameter – any $\frac{k}{r} \in (0.0, 1.0)$ offers a performance improvement over Iso-C while the performance for $\frac{k}{r} \in [0.5, 0.9]$ varies by less than 0.5% from the optimal one.

## 6. Conclusion

In this work, we introduced an isotropic model merging framework that enhances alignment between task-specific and merged model subspaces to significantly improve the multi-task performance of the final merged model. We proposed Iso-C, which leverages Singular Value Decomposition to equalize singular values and create a more balanced representation across tasks, and Iso-CTS, which further incorporates task-specific directions to retain unique task features while preserving shared knowledge. Iso-CTS achieves state-of-the-art results across multiple model scales and task sets, demonstrating that subspace alignment is a critical factor in effective model merging. These findings provide new insights into model merging and pave the way for the future development of more effective techniques to combine the knowledge of multiple models.

**Limitations.** The common subspace is determined by Task Arithmetic, which can be suboptimal, and better methods could be developed. Although the proposed methods achieve state-of-the-art results in the LoRA merging scenario, they could be adapted to leverage the low-rank structure of task matrices to further improve the performance and efficiency.

## Acknowledgements

Daniel Marczak is supported by National Centre of Science (NCN, Poland) Grant No. 2021/43/O/ST6/02482. This work was supported by Horizon Europe Programme under GA no. 101120237, project "ELIAS: European Lighthouse of AI for Sustainability". Simone Magistri acknowledges travel support from ELIAS (GA no 101120237). We acknowledge the Spanish project PID2022-143257NB-I00, financed by MCIN/AEI/10.13039/501100011033 and FEDER, and Funded by the European Union ELLIOT project. Bartłomiej Twardowski acknowledges the grant RYC2021-032765-I and National Centre of Science (NCN, Poland) Grant No. 2023/51/D/ST6/02846. Andrew D. Bagdanov acknowledges funding support from the Italian national project "Collaborative Explainable neuro-symbolic AI for Decision Support Assistant", CAI4DSA, CUP B13C23005640006.

## Impact Statement

This paper aims to advance the field of Machine Learning, specifically the subfield focused on merging models fine-tuned on different tasks to create a more effective multi-task model. With the growing popularity of deep learning, increasingly powerful open-source models are becoming widely available and are being adopted in both research and industry. Advances in model merging could enhance the flexibility of utilizing these models by providing an efficient way to combine their specialized capabilities. Beyond this, our paper presents work whose goal is to advance the field of Machine Learning. There are many potential societal consequences of our work, none of which we feel must be specifically highlighted here.

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

# A. Theoretical properties of `Iso-C`

In this Appendix, we discuss the theoretical properties of `Iso-C` by explicitly showing the connection between spectral skewness and the increased subspace dimensionality $k_M$ in the merged model achieved by `Iso-C`, which leads to a higher Subspace Alignment Ratio (SAR). Moreover, we explain why the increased SAR reduces inter-task interference. Finally, we highlight the limitations of `Iso-C` that lead to the development of `Iso-CTS`.

## A.1. Spectral skewness and the definition of $k_M$

In this Section, we show that the number of dominant components $k_M$ of the merged model $\Delta_M$ (see Equation (6)) is directly influenced by the skewness of its singular value spectrum. Using the singular value decomposition (SVD), let $\Delta_M = U\Sigma V^T$, where $\Sigma = \text{diag}(\sigma_1, \ldots, \sigma_r)$. By the definition of Frobenius norm:

$$\|\Delta_M\|_F^2 = \sum_{i=1}^{r} \sigma_i^2, \quad \|\Delta_M - \Pi_{k,M}\Delta_M\|_F^2 = \sum_{i=k+1}^{r} \sigma_i^2.$$

Hence, the relative approximation error becomes:

$$\frac{\|\Delta_M - \Pi_{k,M}\Delta_M\|_F^2}{\|\Delta_M\|_F^2} = \frac{\sum_{i=k+1}^{r} \sigma_i^2}{\sum_{i=1}^{r} \sigma_i^2}.$$

Accordingly, $k_M$ can be defined in terms of singular values:

$$k_M = \min\left\{ k : \frac{\sum_{i=k+1}^{r} \sigma_i^2}{\sum_{i=1}^{r} \sigma_i^2} \leq \epsilon^2 \right\}.$$

This formulation explicitly shows how the skewness of the spectrum $\{\sigma_i\}$ controls $k_M$. When $\Delta_M$ has a skewed spectrum (e.g. $\sigma_1^2 \gg \sum_{i=2}^{r} \sigma_i^2$), a small $k_M$ is sufficient to meet the error bound. This explains why Task Arithmetic $\Delta_{TA}$ ($\beta = 0$ in Figure 4a) – which has a skewed spectrum – yields a smaller $k_{TA}$ than `Iso-C`, whose flatter spectrum leads to a larger $k_{Iso-C}$. Therefore, expressing $k_M$ directly in terms of singular values highlights the link between the spectral skewness and subspace dimensionality.

## A.2. `Iso-C` increases Subspace Alignment Ratio (SAR)

In this Section, we formally show how `Iso-C` increases Subspace Alignment Ratio (SAR) by expanding the effective subspace dimensionality of the merged model – from $k_{TA}$ in Task Arithmetic to $k_{Iso-C}$ in `Iso-C`.

The rank $k_M$ defines the effective rank of the subspace identified by the merged model and it is directly determined directly by its spectrum (as discussed Appendix A.1). Let $k_{TA}$ be the effective rank of $\Delta_{TA}$, and define

$$T = \{u_1, .., u_{k_{TA}}\}$$

as the orthonormal basis formed by those $k_{TA}$ singular vectors. Flattening the spectrum of $\Delta_{TA}$ (Figure 4a), yields $\Delta_{Iso-C}$ with effective rank $k_{Iso-C} > k_{TA}$ (as discussed in Appendix A.1). This flattening modifies only the singular values of TA, leaving the singular vectors unchanged. Therefore, the original subspace $T$ is contained within the larger subspace spanned by the top singular vectors of $\Delta_{Iso-C}$, defined as:

$$I = \{u_1, .., u_{k_{TA}}, .., u_{k_{Iso-C}}\}.$$

Thus, by construction, we have $T \subset I$.

For simplicity, let $\Pi_T = \Pi_{k_{TA}, TA}$ and $\Pi_I = \Pi_{k_{Iso-C}, Iso-C}$ denote the projection operators onto the subspaces spanned by $T$ and $I$, respectively. Since $T \subset I$, for any matrix $\Delta_t$ it holds that:

$$\text{SAR}(\Delta_t, \Delta_{TA}; k_{TA}) = \frac{\|\Pi_T \Delta_t\|_F}{\|\Delta_t\|_F} \leq \frac{\|\Pi_I \Delta_t\|_F}{\|\Delta_t\|_F} = \text{SAR}(\Delta_t, \Delta_{Iso-C}; k_{Iso-C}), \tag{16}$$

This inequality holds because by definition:

$$\frac{||\Pi_T \Delta_t||_F^2}{||\Delta_t||_F^2} = \frac{\sum_{i=1}^{k_{\mathrm{TA}}} \sum_j \langle u_i, \Delta_t^{(j)} \rangle^2}{||\Delta_t||_F^2} \leq \frac{\sum_{i=1}^{k_{\mathrm{TA}}} \sum_j \langle u_i, \Delta_t^{(j)} \rangle^2 + \sum_{i=k_{\mathrm{TA}}+1}^{k_{\mathrm{Iso\text{-}C}}} \sum_j \langle u_i, \Delta_t^{(j)} \rangle^2}{||\Delta_t||_F^2} = \frac{||\Pi_I \Delta_t||_F^2}{||\Delta_t||_F^2},$$

where $\Delta_t^{(j)}$ denotes the $j$-th column of $\Delta_t$.

The equality in Equation (16) holds only if the additional vectors added to the basis $T$ – that is $\{u_{k_{\mathrm{TA}}+1}, \ldots, u_{k_{\mathrm{Iso\text{-}C}}}\}$ – are orthogonal to each $\Delta_t^{(j)}$ or, equivalently, if they do not intersect the column space of $\Delta_t$ (i.e. its left singular vectors).

Hence, in general a lower $k_{\mathrm{M}}$ yields smaller or equal SAR than a larger $k_{\mathrm{M}}$. However, our empirical findings show that enriching the basis $T$ with singular vectors corresponding to smaller singular values in original task arithmetic spectrum (i.e. $\{u_{k_{\mathrm{TA}}+1}, \ldots, u_{k_{\mathrm{Iso\text{-}C}}}\}$) consistently increases the alignment ratio (Figure 4b), implying that these vectors are relevant for representing each task matrix $\Delta_t$ and not orthogonal to its left singular vectors.

This analysis formally supports the claim that increasing the effective rank $k_{\mathrm{M}}$ of the merged matrix – achieved by spectrum flattening in `Iso-C` – leads to a higher Subspace Alignment Ratio.

### A.3. `Iso-C` mitigates inter-task interference

`Iso-C` increases the Subspace Alignment Ratio (SAR), which quantifies how well the principal directions of a task matrix align with the principal directions of the merged model. In this Section, we demonstrate how a higher SAR contributes to mitigate inter-task interference by analyzing the relationship between subspace alignment and changes in internal activations following merging. Specifically, we define the interference as the degradation in a task's internal representation due to merging —- that is, the deviation between the activations of the merged model and those of the corresponding single-task fine-tuned model. Intuitively, we can minimize the task interference by ensuring that the internal representations of task $j$ remain stable after merging.

Let $\theta_0$ be the pre-trained weights for a layer $l$. Define the task matrix $\Delta_j = \theta_j - \theta_0$ and the merged task matrix $\Delta_{\mathrm{M}}$ for the layer $l$. Then, for an input $x_j^{(l)}$, we desire that the post-merging activation $h_j^{(l)} = (\theta_0 + \alpha \Delta_{\mathrm{M}}) x_j^{(l)}$, with $\alpha$ chosen on a validation set, be close to the task-specific activation $\hat{h}_j^{(l)} = (\theta_0 + \Delta_j) x_j^{(l)}$. Hence, we can quantify the interference as:

$$||\hat{h}_j^{(l)} - h_j^{(l)}|| = ||(\Delta_j - \alpha \Delta_{\mathrm{M}}) x_j^{(l)}|| \leq ||\Delta_j - \alpha \Delta_{\mathrm{M}}|| \cdot ||x_j^{(l)}||. \tag{17}$$

To show that the interference is lower when the Subspace Alignment Ratio (SAR) between $\Delta_j$ and $\Delta_{\mathrm{M}}$ is higher, we decompose $\Delta_j$ into components aligned with and orthogonal to $\Delta_{\mathrm{M}}$:

$$\Delta_j = \Delta_j^{||} + \Delta_j^{\perp} \quad \text{where} \quad \Delta_j^{||} = \Pi_{k_{\mathrm{M}}, \mathrm{M}} \Delta_j, \quad \Delta_j^{\perp} = (I - \Pi_{k_{\mathrm{M}}, \mathrm{M}}) \Delta_j, \tag{18}$$

and $\Pi_{k_{\mathrm{M}}, \mathrm{M}}$ is the projection matrix onto the subspace spanned by the top $k_{\mathrm{M}}$ left-singular vectors of $\Delta_{\mathrm{M}}$ (see Equation (6) for the definition of $k_{\mathrm{M}}$). The Subspace Alignment Ratio is then:

$$\mathrm{SAR}(\Delta_j, \Delta_{\mathrm{M}}; k_{\mathrm{M}}) = \frac{||\Pi_{k_{\mathrm{M}}, \mathrm{M}} \Delta_j||_F}{||\Delta_j||_F} = \frac{||\Delta_j^{||}||_F}{||\Delta_j^{||} + \Delta_j^{\perp}||_F}. \tag{19}$$

Similarly, decomposing $\Delta_{\mathrm{M}}$ into $\Delta_{\mathrm{M}}^{||}$ and $\Delta_{\mathrm{M}}^{\perp}$ and substituting Equation (18) in Equation (17), the interference becomes:

$$||\Delta_j - \alpha \Delta_{\mathrm{M}}|| = ||\Delta_j^{||} - \alpha \Delta_{\mathrm{M}}^{||} + \Delta_j^{\perp} - \alpha \Delta_{\mathrm{M}}^{\perp}|| \approx ||\Delta_j^{||} - \alpha \Delta_{\mathrm{M}}^{||} + \Delta_j^{\perp}||, \tag{20}$$

since $k_{\mathrm{M}}$ minimizes the approximation error of $\Delta_{\mathrm{M}}$, leading to $||\Delta_{\mathrm{M}}^{\perp}|| \approx 0$.

If the SAR defined in Equation (19) is close to 1, then $||\Delta_j^{\perp}||$ is small, so the interference in Equation (20) mainly depends on $||\Delta_j^{||} - \alpha \Delta_M^{||}||$. Conversely, if SAR is near zero, the large orthogonal component $\Delta_j^{\perp}$ increases the overall interference, regardless of the choice of $\alpha$. Even with an optimal $\alpha$ chosen via validation, interference cannot be reduced below the norm of the orthogonal component.

`Iso-C` increases the SAR of $\Delta_t$ with the merged model — bringing it close to 1, as shown in the paper — by flattening the singular values. Thus, the optimal $\alpha$ can adjust the merged model such that interference is minimized. In contrast, Task Arithmetic (TA), with SAR varying across tasks, exhibits interference that cannot be reduced below the norm of the orthogonal component. We experimentally evaluate that the interference is lower for `Iso-C` than TA in Appendix D.2.

### A.4. Limitations of `Iso-C` that motivate `Iso-CTS`

This Section details the limitations of `Iso-C` that motivate the development of `Iso-CTS`. Specifically, `Iso-C` relies on the singular vectors obtained through Task Arithmetic to perform model merging. As a result, it tends to underrepresent tasks whose dominant directions have lower intensity compared to the majority, particularly when those directions are orthogonal to the shared (common) directions. This limitation becomes increasingly pronounced as the number and diversity of tasks increase (see Section 4.2).

To make this limitation explicit, we formalize the computation – via SVD – of the first left singular vector in Task Arithmetic, used by `Iso-C`, as the variance maximization problem:

$$u_1 = \arg \max_{||u||=1} ||\Delta_{\text{TA}}^\top u||^2 = u^\top \left( \sum_{t=1}^{T} \Delta_t \Delta_t^\top \right) u + u^\top ( \sum_{t,s=1,t \neq s}^{T} \Delta_t \Delta_s^\top ) u$$

If a particular task $\Delta_j$ has dominant directions with significantly lower intensity compared to the other tasks (i.e. lower Frobenius Norm), then its individual contributions $\Delta_j \Delta_j^\top$ to the total variance becomes smaller. Similarly, cross terms involving $\Delta_j$ will also be comparatively small. Therefore, task $j$ explicitly contributes less to the maximized variance captured by the first principal singular direction.

Moreover, if the directions of $\Delta_j$ are orthogonal or nearly orthogonal to $u_1$, (i.e. $u_1^\top \Delta_j = 0$), task $j$ contributes minimally or not at all along this principal direction. Similar considerations apply to subsequent singular vectors $u_2, \ldots u_k$, defining the common subspace. Finally, as the number of tasks $T$ increases and tasks become more diverse, it becomes increasingly likely that tasks with distinct but smaller-magnitude directions will be underrepresented or absent in the dominant singular directions identified by the task arithmetic decomposition.

The goal of `Iso-CTS` is to address this limitation by incorporating orthogonal directions that are overlooked by the Task Arithmetic spectrum. This strategy yields the greatest improvements in settings with a large number of diverse tasks, as shown in our experimental results.

## B. Computational complexity analysis

In this Section, we analyze the computational complexity of `Iso-C` and `Iso-CTS` and compare it with that of our main competitor, TSV-M (Gargiulo et al., 2025).

Let $\Delta_t \in \mathbb{R}^{n \times n}$, and let $T$ and $L$ be the number of tasks and network layers, respectively. For simplicity, assume that each layer consists of a single squared $n \times n$ matrix.

In our analysis, we focus on the number of SVD performed by each algorithm, as this is by far the most costly component of each algorithm. The complexity of a single SVD on $\Delta_t \in \mathbb{R}^{n \times n}$ is equal to $\mathcal{O}(n^3)$ (Vasudevan & Ramakrishna, 2017). Below, we detail the total computational complexity for each merging method:

- `Iso-C` performs a single SVD on $\Delta_{\text{TA}}$ per layer, with total complexity:

$$\mathcal{O}(\texttt{Iso-C}) = \mathcal{O}(Ln^3)$$

- `Iso-CTS` performs:
    - One SVD on $\Delta_{\text{TA}}$ per layer (lines 2-3, Algorithm 2) with complexity $\mathcal{O}(Ln^3)$
    - One SVD on each $\Delta_t$, for all $T$ tasks and each of the $L$ layers (line 5, Algorithm 2), with complexity $\mathcal{O}(TLn^3)$
    - Two SVDs on two matrices $U_*, V_* \in \mathbb{R}^{n \times n}$ per layer (line 11, Algorithm 2), with complexity of $\mathcal{O}(2Ln^3)$.

    Therefore, the total complexity equals:

$$\mathcal{O}(\texttt{Iso-CTS}) = \mathcal{O}(Ln^3 + TLn^3 + 2Ln^3) = \mathcal{O}((T+3)Ln^3) = \mathcal{O}(TLn^3)$$

- TSV-M (Gargiulo et al., 2025) performs:
    - $T$ SVDs per layer on each task matrix (line 1, Alg. 1 from Gargiulo et al. (2025)): $\mathcal{O}(TLn^3)$
    - Two additional SVDs per layer (lines 10-11, Alg.1 from Gargiulo et al. (2025)): $\mathcal{O}(2Ln^3)$

Yielding the total complexity:

$$\mathcal{O}(\text{TSV}) = \mathcal{O}(TLn^3 + 2Ln^3) = \mathcal{O}((T+2)Ln^3) = \mathcal{O}(TLn^3)$$

While `Iso-CTS` and TSV-M share the same asymptotic complexity, `Iso-CTS` incurs slightly more overhead due to the SVD on $\Delta_{\text{TA}}$ (lines 2-3, Algorithm 2). Both methods can be further optimized by computing Truncated SVDs for `Iso-CTS` and TSV-M, since only a few components are retained. This reduces the complexity for both approaches. `Iso-C` is the most computationally efficient algorithm – its complexity is constant with respect to number of task $T$.

## C. Experimental details

In this Appendix, we provide the dataset and implementation details used to carry out the experiments presented in the paper.

### C.1. Datasets

The 8-dataset benchmark consists of: Cars (Krause et al., 2013), DTD (Cimpoi et al., 2014), EuroSAT (Helber et al., 2019), GTSRB (Stallkamp et al., 2011), MNIST (Lecun et al., 1998), RESISC45 (Cheng et al., 2017), SUN397 (Xiao et al., 2016), and SVHN (Netzer et al., 2011).

The 14-dataset benchmark builds on the preceding one, incorporating six additional datasets: CIFAR100 (Krizhevsky & Hinton, 2009), STL10 (Coates et al., 2011), Flowers102 (Nilsback & Zisserman, 2008), OxfordIIITPet (Parkhi et al., 2012), PCAM (Veeling et al., 2018), and FER2013 (Goodfellow et al., 2013).

Finally, the 20-dataset benchmark includes the preceding 14 plus the following six: EMNIST (Cohen et al., 2017), CIFAR10 (Krizhevsky & Hinton, 2009), Food101 (Bossard et al., 2014), FashionMNIST (Xiao et al., 2017), RenderedSST2 (Socher et al., 2013), and KMNIST (Clanuwat et al., 2018).

### C.2. Implementation details

Our method relies on SVD, which is defined for two-dimensional matrices $\Delta \in \mathbb{R}^{m \times n}$. However, some weights of the neural networks are represented by vectors $\delta \in \mathbb{R}^n$, e.g. bias vectors and parameters of layer normalization (Ba et al., 2016). Therefore, following Gargiulo et al. (2025), we apply simple averaging to combine these parameters.

## D. Additional experiments

In this Appendix, we present additional experiments that complement the main paper, including comparisons with new vision baselines using Task Arithmetic model checkpoints (Ilharco et al., 2023) for evaluation. We empirically assess the reduced interference of `Iso-C` compared to Task Arithmetic and analyze the impact of the scaling factor $\alpha$ on our approaches. Finally, we present an ablation study showing what happens when spectrum flattening is applied to each task model individually.

### D.1. Additional vision baselines

In this Section we provide results with additional methods: Fisher Merging (Matena & Raffel, 2021), RegMean (Jin et al., 2023), PCB (Du et al., 2024), MaTS (Tam et al., 2023) and CART (Lee et al., 2025). These methods were originally evaluated on checkpoints from Task Arithmetic (Ilharco et al., 2023) provided for 8 tasks on ViT-B/32 and ViT-L/14. We follow this experimental protocol with `Iso-C` and `Iso-CTS` and present the results in Table 4. `Iso-CTS` sill achieves state-of-the-art performance followed by `Iso-C`. Note that the results differ from Table 1 where we used checkpoints from Consensus Merging (Wang et al., 2024b).

*Table 4.* Additional baselines for merging ViT-B/32 and ViT-L/14 on 8 tasks. We report absolute accuracy.

| Method | ViT-B/32 | ViT-L/14 |
|---|---|---|
| Zero-shot | 48.3 | 64.7 |
| Fine-tuned | 90.5 | 94.2 |
| Task Arithmetic | 70.5 | 84.6 |
| Fisher Merging | 68.3 | 83.7 |
| RegMean | 71.8 | 82.2 |
| PCB | 76.3 | 87.5 |
| MaTS | 82.6 | 90.2 |
| CART | 83.0 | 90.8 |
| `Iso-C` (Ours) | 84.1 | 92.5 |
| `Iso-CTS` (Ours) | **84.3** | **93.0** |

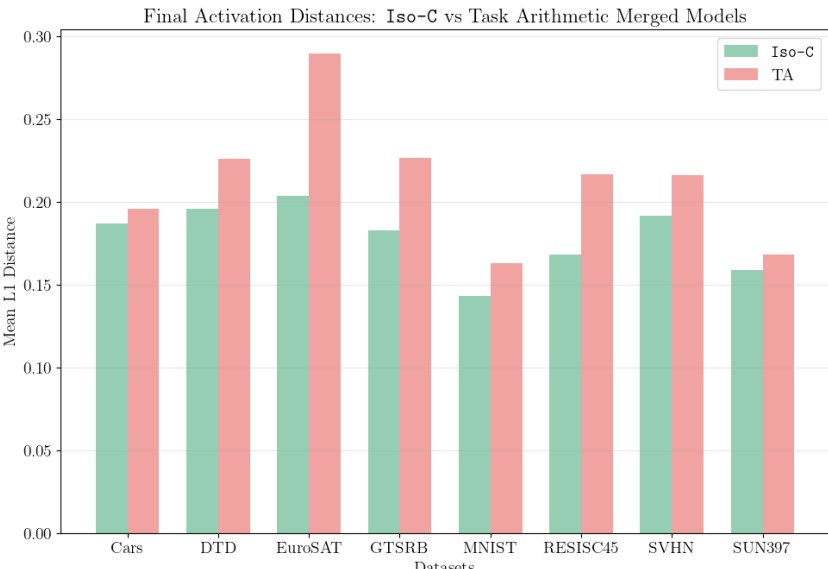

*Figure 7.* Mean L1 distance between the final embeddings of task-specific models and the merged one for `Iso-C` and TA. We used ViT-B/16 model.

### D.2. Interference quantification

In this Section, we experimentally show that merging interference (defined in Appendix A.3) is lower when merging is performed with `Iso-C` than with TA. Following Yang et al. (2024), we measure the interference as L1 distance between the final embeddings of task-specific models and merged one. In Figure 7 we present the results for merging 8 tasks on ViT-B/16. We observe that the interference is lower for `Iso-C` than for TA highlighting the effectiveness of `Iso-C` in reducing interference when merging models.

### D.3. Selection of scaling coefficient $\alpha$

In Figure 10, we present the relationship between the validation accuracy and scaling factor $\alpha$. We observe that TA is very sensitive to the selection of $\alpha$, which potentially may require a more fine-grained search. On the other hand, both `Iso-C` and `Iso-CTS` are more robust to $\alpha$ selection, resembling the task-specific models. For reproducibility, In Table 5, we provide the optimal $\alpha$ value chosen on the held-out validation set for each model and number of tasks.

### D.4. Importance of isotropic scaling in `Iso-CTS`

In this Section we ablate the need for isotropic scaling in `Iso-CTS`. We present the comparison of the performance of `Iso-CTS` with and without isotropic scaling (Equation 13) in Figure 8. We observe that isotropic scaling is indeed a

*Table 5.* Optimal $\alpha$ value chosen on a held-out validation set for different model types and numbers of tasks for `Iso-C` and `Iso-CTS`.

| Method | Model | 8 tasks | 14 tasks | 20 tasks |
|---|---|---|---|---|
| Iso-C | ViT/32-B | 1.30 | 1.00 | 0.90 |
| | ViT/16-B | 1.40 | 1.00 | 0.80 |
| | ViT/14-L | 1.50 | 1.30 | 1.00 |
| Iso-CTS | ViT/32-B | 1.50 | 1.20 | 1.10 |
| | ViT/16-B | 1.60 | 1.20 | 1.10 |
| | ViT/14-L | 1.90 | 1.50 | 1.20 |

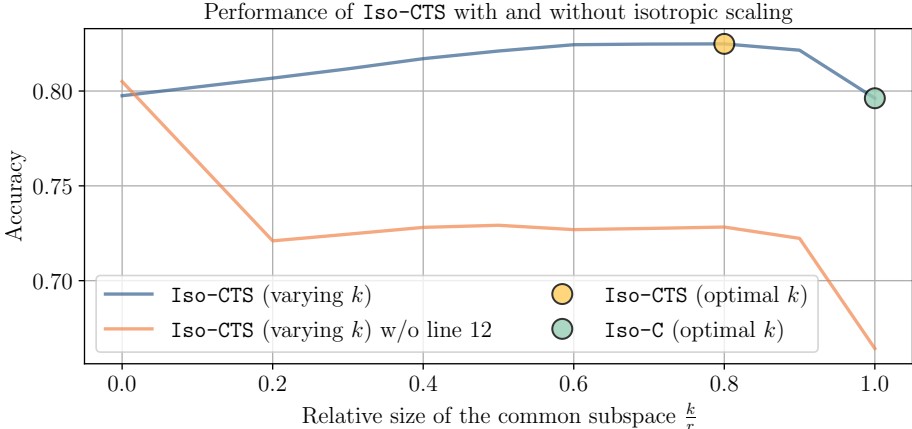

*Figure 8.* Performance of `Iso-CTS` with and without isotropic scaling (Eq.13). Isotropic scaling is a crucial component of `Iso-CTS`. Results for merging 20 tasks with ViT-B/16.

crucial component of `Iso-CTS` as long as common subspace exists. When only task-specific subspaces are in use ($\frac{k}{r} = 0$), isotropic scaling does not make a significant difference. However, the design in Algorithm 2 also plays an important role, especially when the number of merged models increases, leading to up to 2.8% improvement over `Iso-C` on 20 tasks (see Table 1).

### D.5. Applying `Iso` to individual task matrices

Flattening the skewed spectrum of singular values significantly improves the performance of the merged model, as demonstrated in Section 5.4. One may wonder if this operation might also be an effective strategy for improving single-task models. Figure 11 presents the performance of task-specific models in their original form along with their modified versions with singular value spectra of their task matrices flattened (which is equivalent to performing `Iso-C` for a single model). We observe a 3.3% drop in average performance across tasks. Therefore, the reason for the success of `Iso-C` lies in its ability to mitigate the negative effects of summing task matrices, not in inadvertently improving the original individual task matrices.

## E. Additional visualizations

In this Appendix, we provide additional visualizations that could not be included in the main paper due to space constraints. These include the spectra of task matrices, the Subspace Alignment Ratio per layer, and the correlation between Normalized Accuracy Improvement and Subspace Alignment Ratio when using the larger ViT-L/14 model.

### E.1. Visualization of task matrix spectra

When visualizing spectra of singular values of task matrices (Figure 1 and Figure 4a), we selected an output projection matrix $W^O$ from layer $\ell = 4$ of ViT/B-16 as an illustrative example. In Figure 12, we present spectra across a variety of layers of ViT/B-16 for the task matrices of task-specific models, TA, `Iso-C` and `Iso-CTS`.

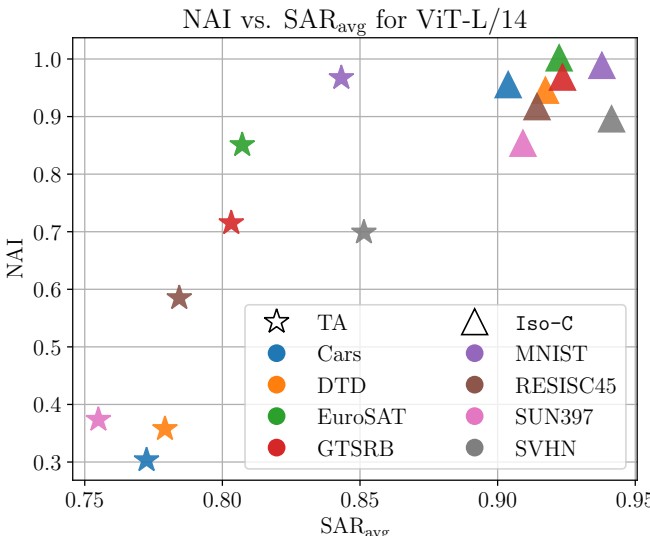

*Figure 9.* Normalized Accuracy Improvement (NAI) vs. Average Subspace Alignment Ratio (SAR$_{avg}$) for ViT-L/14.

### E.2. Visualization of per layer Subspace Alignment Ratio

In Figure 3a and Figure 4b in the main paper, we presented SAR$_{avg}$ – Subspace Alignment Ratio averaged across all the task matrices. In this Section, we present SAR at different depths of the model. Specifically, we calculate SAR between fine-tuned and merged weight matrices and an average of all the matrices for a given layer of the ViT-B/16 model. We present the results in Figure 13. We observe that the alignment is higher for `Iso-C` across all layers of the vision transformer. One may expect early layers to be more aligned but we find that for both approaches the alignment is similar across the layers.

### E.3. Visualization of Normalized Accuracy Improvement versus Subspace Alignment Ratio for ViT-L/14

In Figure 9 we replicate the experiment from Figure 3a (conducted on ViT-B/16) on ViT-L/14. The observations from the main paper hold – Normalized Accuracy Improvement strongly correlates with average Subspace Alignment Ratio, and increasing SAR$_{avg}$ via merging with `Iso-C` leads to better performance.

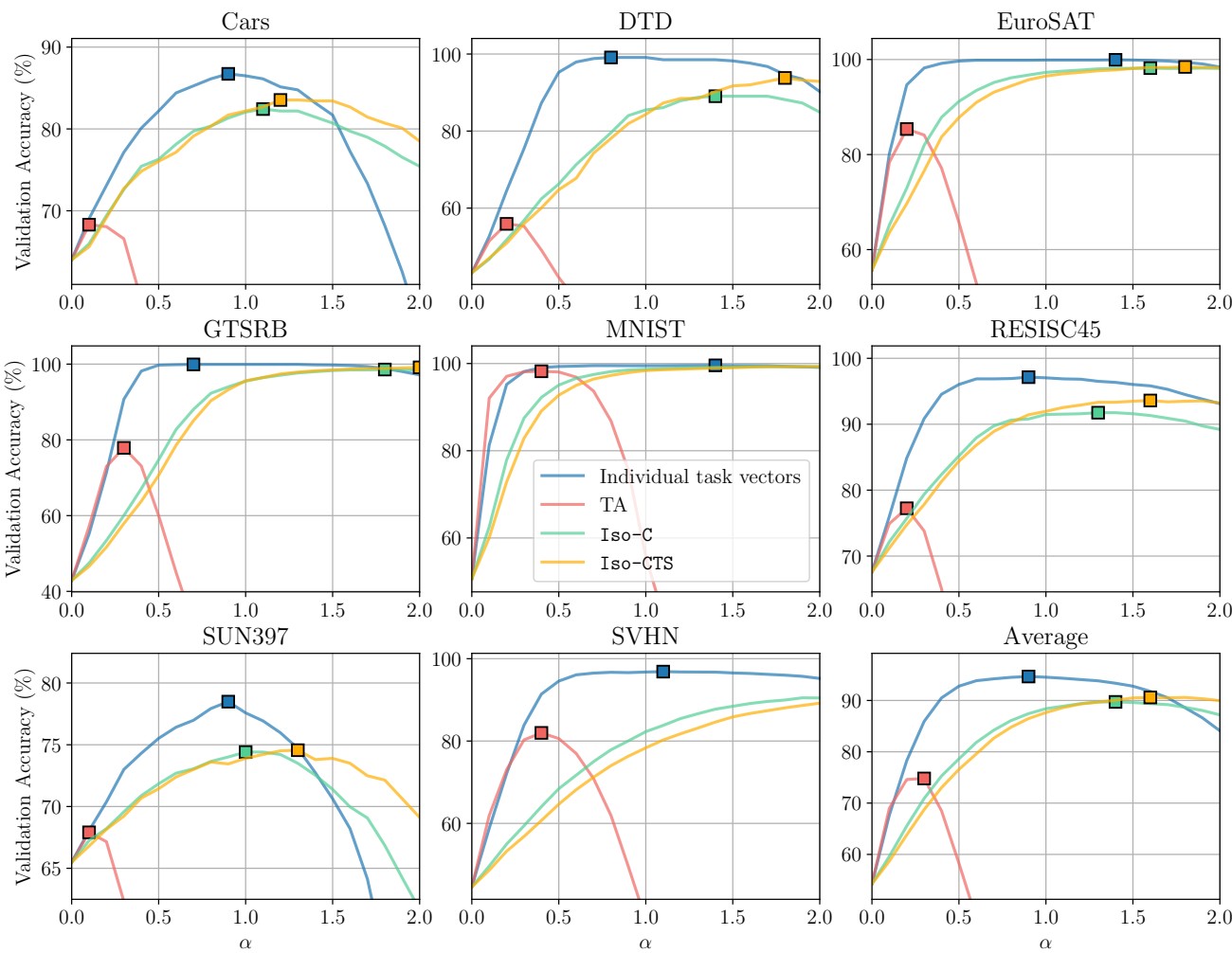

*Figure 10.* TA is sensitive to the selection of $\alpha$, while both `Iso-C` and `Iso-CTS` are more robust to $\alpha$ selection, resembling the task-specific models. The $\alpha$ is chosen based on the best average performance on the validation set across tasks. The bottom right subplot denotes the optimal $\alpha$ for each method (Eq. (3), Eq. (9) and Eq. (14)). The model is ViT-B/16.

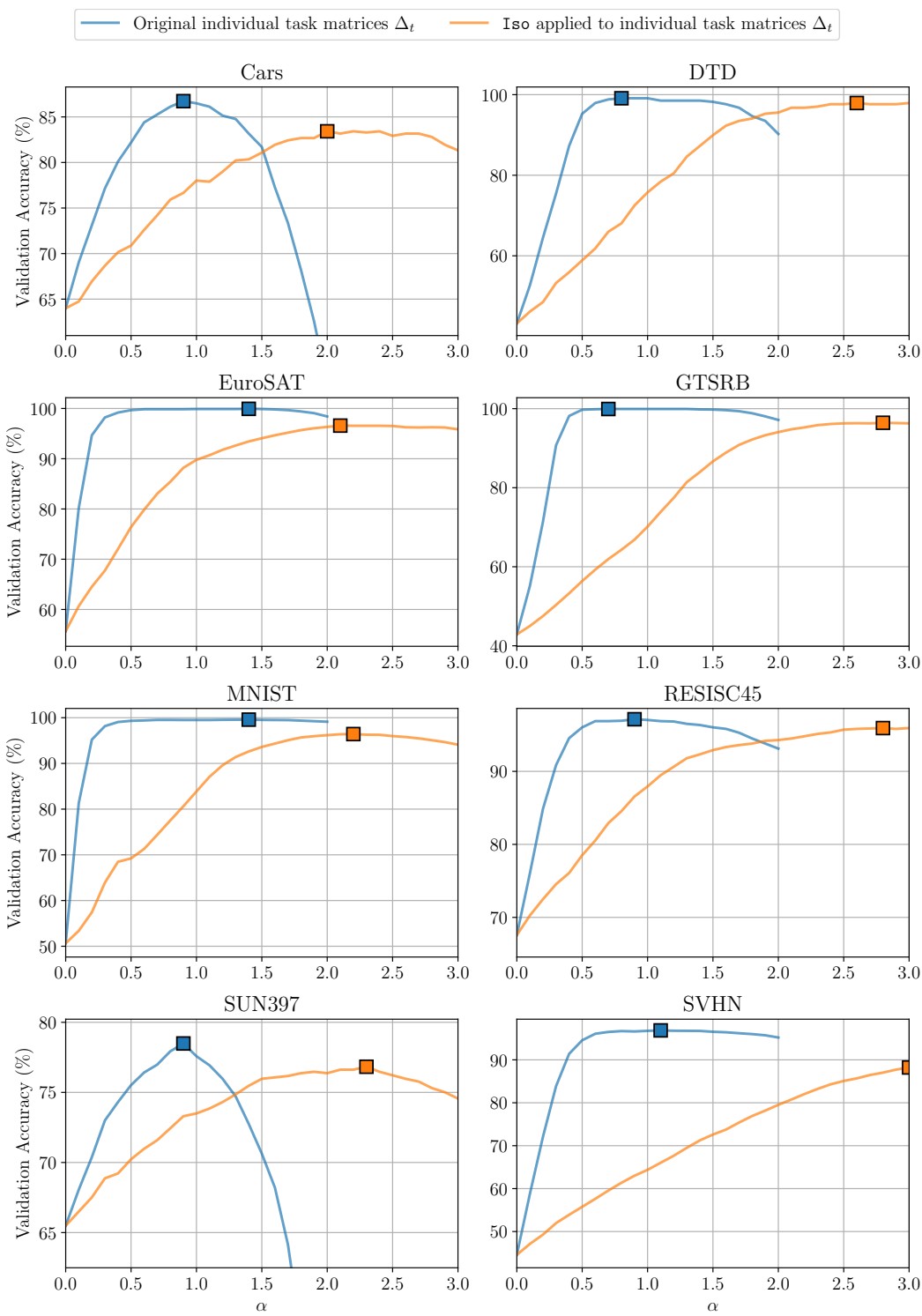

*Figure 11.* Validation Accuracy while scaling task matrices with $\alpha$ coefficient (Eq. (3) applied for a single task). We observe a performance gap between the accuracy of original and modified models for the optimal values of $\alpha$ (denoted by square).

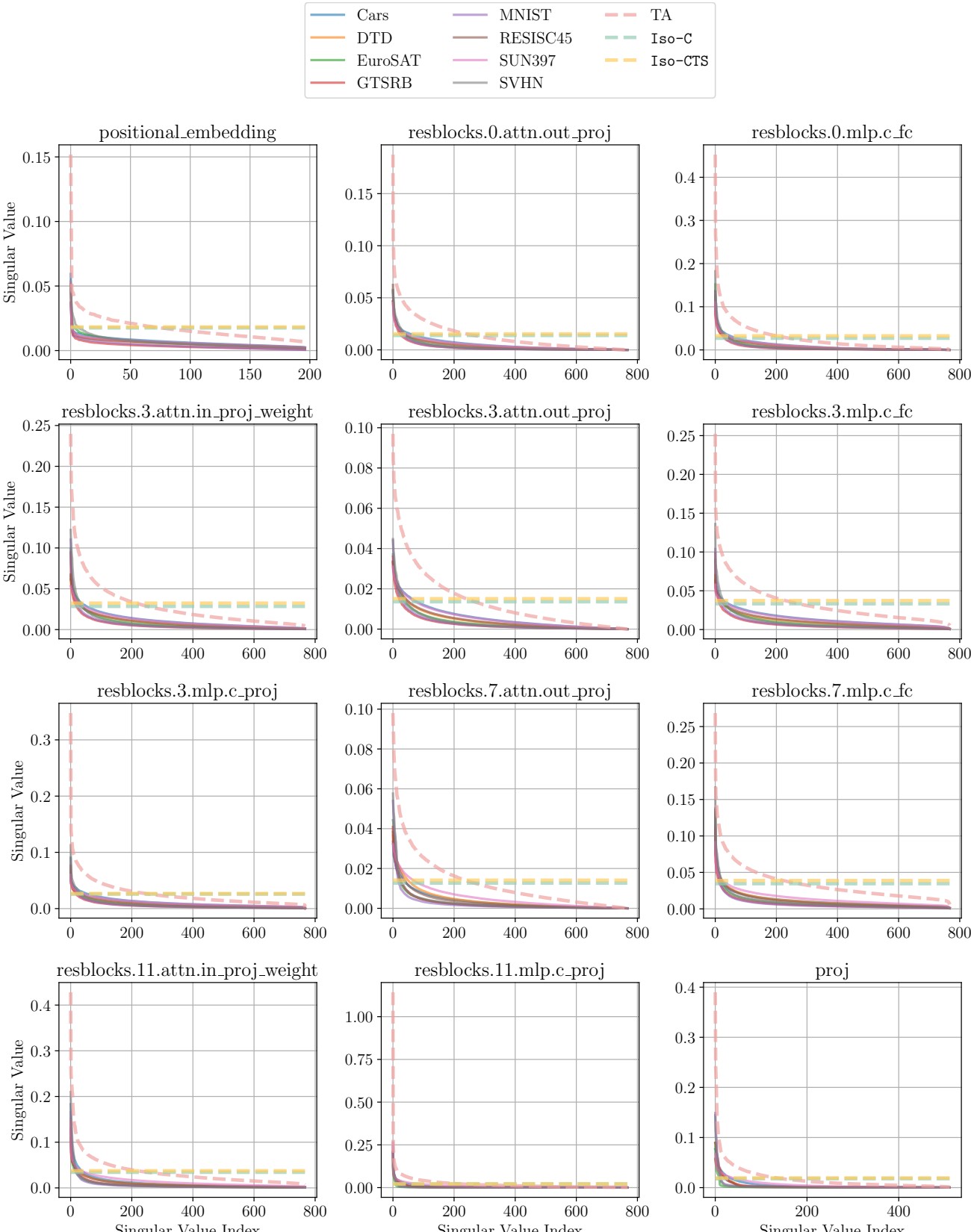

*Figure 12.* Visualization of singular value spectra of different task matrices for different types of layers in ViT/B-16.

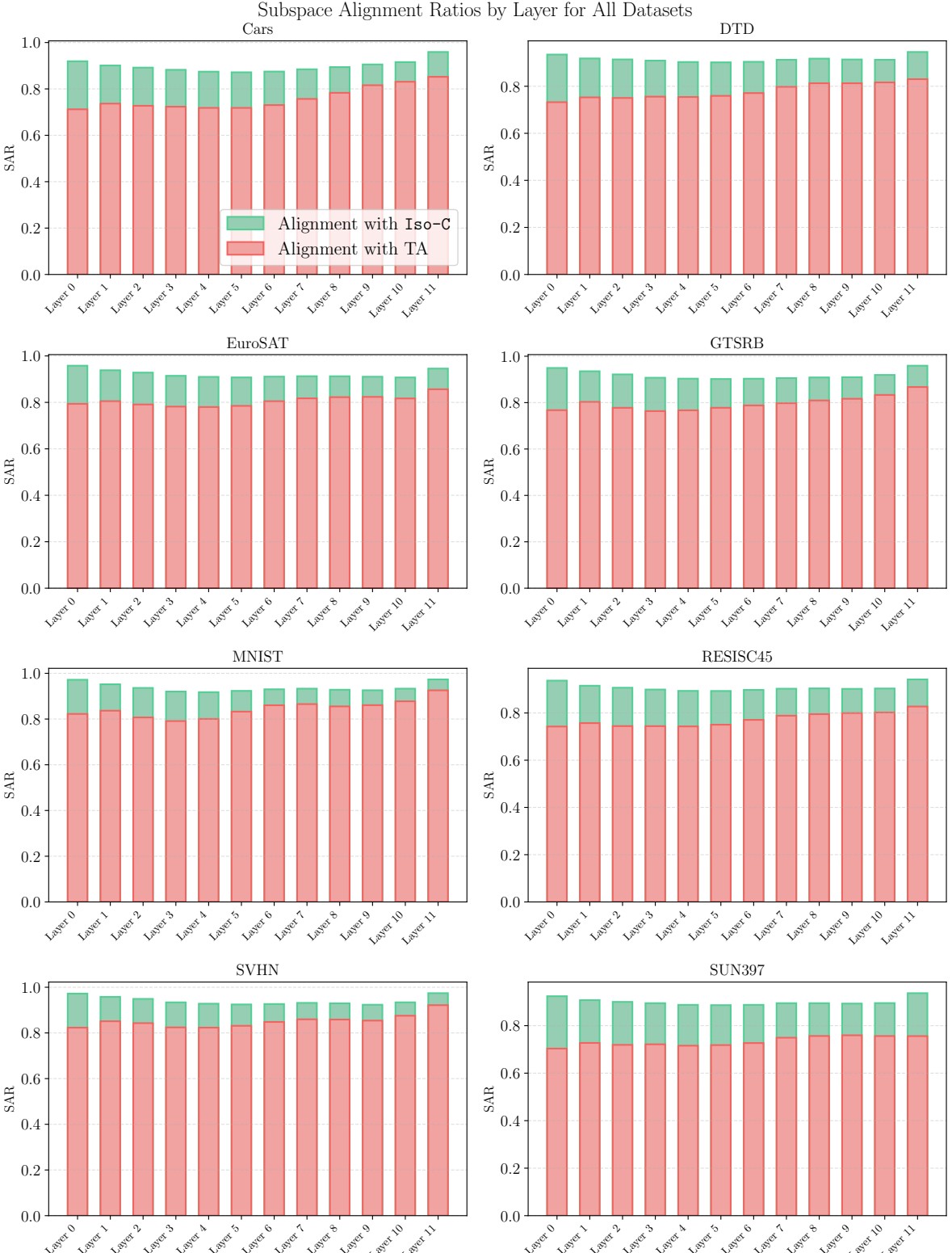

*Figure 13.* Per layer Subspace Alignment Ratio between fine-tuned and merged weight matrices for ViT-B/16.

