# OpenReview forum: "No Task Left Behind: Isotropic Model Merging with Common and Task-Specific Subspaces"
_ICML.cc/2025/Conference — ICML 2025 poster_

### Official Review · Reviewer_38Gq · 2025-02-19

**Overall Recommendation:** 4

**Summary:**

This paper demonstrates that alignment between the individual components of task-specific and merged matrices is strongly correlated with performance improvements over a pre-trained model. Building on this finding, the authors propose an isotropic merging framework that flattens the singular value spectrum of task matrices, thereby enhancing alignment and narrowing the performance gap. Furthermore, they incorporate both common and task-specific subspaces to further optimize alignment and boost performance. The proposed approach achieves state-of-the-art results.

**Claims And Evidence:**

The proposed Normalized Accuracy Improvement and Subspace Alignment Ratio are supported by evidence. These metrics provide quantitative validation for the proposed isotropic merging framework, showcasing the alignment improvements and their direct impact on model performance.

**Essential References Not Discussed:**

Section 4.2, which discusses retaining components from the common subspace and the orthogonal projection in Equation 10, bears resemblance to the shared subspace optimization concept in DOGE [2]. It is recommended to discuss this.

[2] Modeling Multi-Task Model Merging as Adaptive Projective Gradient Descent. arXiv.

**Experimental Designs Or Analyses:**

The experimental design followed Task Singular Vectors, but was limited to vision tasks.

**Methods And Evaluation Criteria:**

It is intuitive to keep task-specific knowledge in the near-zero singular values part of the common subspace and discard the unimportant part.

**Other Comments Or Suggestions:**

**Suggestions**: Supplement experiments on NLP tasks to verify the generalizability of the method and make the article more complete.

**Other Strengths And Weaknesses:**

**Strengths**: The article is well-written and clear, with simple and effective methods achieving state-of-the-art results in model merging. The proposed methods and metrics are novel and intuitive.

**Weaknesses**: Lack of further analysis, such as why SVD can be used for model merging due to the redundancy of parameters caused by fine-tuning. It now seems more like an experimental discovery, and these further analyses would elevate the article to a higher level.

**Questions For Authors:**

1. Why can subspace alignment eliminate conflicts and improve performance?
2. How is the ratio of singular values controlled between the common and task-specific subspaces?

**Relation To Broader Scientific Literature:**

The method is a further improvement of Task Singular Vectors, removing noise through SVD decomposition.

[1] Task Singular Vectors: Reducing Task Interference in Model Merging. arXiv.

**Theoretical Claims:**

This paper does not conduct theoretical analysis.

---

> ### Author Rebuttal · Authors · 2025-03-31
>
> We are pleased that the Reviewer appreciates the soundness of our introduced metrics, the simplicity and effectiveness of the proposed approaches, and clear writing. We thank the Reviewer for the comments and we respond below to specific points.
>
> >[Reference 1 (R1)]: *Section 4.2, which discusses retaining components from the common subspace and the orthogonal projection in Equation 10, bears resemblance to the shared subspace optimization concept in DOGE [2]. It is recommended to discuss this.*
>
> Thank you for pointing us to this recent reference. We were not aware of DOGE paper at the time of preparing a submission as the preprint appeared two weeks before the deadline. Here we compare this approach with ours:
>
> - *Definition of common/shared subspace:* We define the common subspace as top-k components from sum of individual task matrices. DOGE defines shared as concatenation of top-k components from each task matrix followed by SVD, which resembles the TSV method.
> - *Orthogonal projection:* Both Iso-CTS and DOGE use an idea of orthogonal projection. DOGE uses it on the gradient of $\Delta$ to restrict the optimization process from changing the shared space. Iso-CTS uses the orthogonal projection on the level of weight matrices to determine the task-specific subspace that is orthogonal to the common subspace.
>
> Iso-CTS and DOGE both use ideas of common/shared subspace and orthogonal projection in different ways. Moreover, the results of our approaches are better than DOGE. We believe that this discussion is very significant and we will add a detailed version of it to the revised manuscript.
>
> >[Weakness 1 (W1)]: *Lack of further analysis, such as why SVD can be used for model merging due to the redundancy of parameters caused by fine-tuning. It now seems more like an experimental discovery, and these further analyses would elevate the article to a higher level.*
>
> It is known that fine-tuning of large pre-trained models results in low-rank parameter update. This observation enables efficient fine-tuning of models using inherently low-rank adaptation techniques such as LoRA. Consequently, the recent TSV paper shows how low-rank approximation of the parameter update matrices, obtained using SVD, can be used to facilitate model merging.
>
> In our paper, we propose to extend the scope of SVD-based analysis for the purpose of model merging. Most importantly, by introducing the SAR metric we show that SVD can help in understanding the overlap between task-specific and merged matrices. Moreover, we show that by modifying the spectrum of singular values of merged matrix we can increase the alignment between task and merged matrix (see the Response to Reviewer **ff9S** (section Q2) for detailed discussion).
>
> >*[Suggestion 1 (S1)]: Supplement experiments on NLP tasks to verify the generalizability of the method and make the article more complete.*
>
> We present NLP results in response to Reviewer **mnUL** (section C3). Iso-C and Iso-CTS outperform other baselines across two presented settings.
>
> >[Question 1 (Q1)]: *Why can subspace alignment eliminate conflicts and improve performance?*
>
> Consider the Subspace Alignment Ratio between a task matrix and merged task matrix. SAR quantifies an overlap between the subspaces spanned by dominant singular vectors of these matrices. If the SAR is low, the overlap between the subspaces of these matrices is low and the corresponding singular vectors are close to orthogonal. Therefore, the merged matrix cannot reliably represent the dominant components of task matrix if SAR between them is low. Therefore, low SAR leads to low performance on the corresponding task. Conversely, high SAR indicates high subspace overlap, indicating that the merged matrix can reliably represent the important components of the task matrix and results in high performance.
>
> >[Q2]: *How is the ratio of singular values controlled between the common and task-specific subspaces?*
>
> The ratio of singular values between the common and task-specific subspaces is controlled by the hyperparameter $k$ that is fixed for all the experiments. $k$ is chosen such that $k/r$ for a single layer is equal to 0.8. The final paragraph of Section 5.3 from the paper contains an analysis of impact of $k$ on performance.

---

> > ### Comment · Reviewer_38Gq · 2025-04-02
> >
> > Thank you for the author's rebuttal. Most of the explanations you provided were things I already understood during my review, and I was trying to ask for deeper explanations, such as why subspace alignment eliminates conflicts. You have merely repeated the definition and findings of SAR from the paper. Because the singular value ​​​​average reduces the Frobenius norm and condition number of task vectors, which is why you need to search for a larger $\lambda$, this is unstable on LLM. As I said before, these questions are meant to encourage further analysis to elevate the article to a higher level. I also want to point out that the checkpoints used by ISO and TSV are different from those used in most model merging methods (from Task Arithmetic), which leads to slightly higher results. Additionally, I would like to ask why the authors did not compare with methods such as EMR merging or Twin merging.
> >
> > The current response does not satisfy me. If the further responses are better, I will increase my score.

---

> > > ### Author Response · Authors · 2025-04-07
> > >
> > > >*Why subspace alignment eliminates conflict.*
> > >
> > > We thank the Reviewer for encouraging us to think more deeply about the relationship between subspace alignment and merging conflicts. We will incorporate this analysis in the revised manuscript.
> > >
> > > Intuitively, we can minimize the task interference (i.e. eliminate the conflicts) by ensuring that the internal representations of task $j$ remain stable after merging. Let $\theta_0$ be the pre-trained weights for a layer $l$. Define the task matrix $\Delta_j=\theta_j-\theta_0$ and the merged task matrix $\Delta_M$ for the layer $l$. Then, for an input $x_j^{(l)}$, we desire that the post-merging activation $h_j^{(l)}=(\theta_0+\alpha\Delta_M)x_j^{(l)}$, with $\alpha$ chosen on a validation set, be close to the task-specific activation $\hat{h}_j^{(l)}=(\theta_0+\Delta_j)x_j^{(l)}$. Hence, we can quantify the interference using:
> > >
> > > $$||\hat{h}_j^{(l)}-h_j^{(l)}||=||(\Delta_j-\alpha\Delta_M)x_j^{(l)}||\leq||\Delta_j-\alpha\Delta_M||\cdot||x_j^{(l)}||$$
> > >
> > > To show that the interference is lower when the Subspace Alignment Ratio (SAR) between $\Delta_j$ and $\Delta_M$ is higher, we decompose $\Delta_j$ into components aligned with and orthogonal to $\Delta_M$:
> > >
> > > $$\Delta_j=\Delta_j^{||}+\Delta_j^{\perp}\mbox{ for }\Delta_j^{||}=\Pi_{k_M,M}\Delta_j\mbox{ and }\Delta_j^{\perp}=(I-\Pi_{k_M,M})\Delta_j,$$
> > >
> > > where $\Pi_{k_M,M}$ is the projection matrix onto the subspace spanned by the top $k_M$ left-singular vectors of $\Delta_M$ (see Eqs. 5-6 for their definitions). By rewriting the SAR we have:
> > >
> > > $$SAR(\Delta_j,\Delta_M)=\frac{||\Delta_j^{||}||_F}{||\Delta_j^{||}+\Delta_j^{\perp}||_F}.$$
> > >
> > > Similarly, decomposing $\Delta_M$ in $\Delta_M^{||}$ and $\Delta_M^{\perp}$, we write:
> > >
> > > $$||\Delta_j-\alpha\Delta_M||=||\Delta_j^{||}-\alpha\Delta^{||}_M+\Delta_j^{\perp}-\alpha\Delta^{\perp}_M||\approx||\Delta_j^{||}-\alpha\Delta^{||}_M+\Delta_j^{\perp}||,$$
> > > since $k_M$ minimizes the approximation error of $\Delta_M$ (i.e.$\Delta^{\perp}_M\approx0$).
> > >
> > > If SAR is close to 1, then $||\Delta_j^{\perp}||$ is small, so interference mainly depends on $||\Delta_j^{||}-\alpha\Delta^{||}_M||$. Conversely, if SAR is near zero, the large orthogonal component $\Delta_j^{\perp}$ increases the overall interference, regardless of the choice of $\alpha$. Even with optimal $\alpha$ chosen via validation, **interference cannot be reduced below the norm of the orthogonal component**.
> > >
> > > Iso-C increases SAR of $\Delta_t$ with the merged model — bringing it close to 1, as shown in the paper — by flattening the singular values. Thus, the optimal $\alpha$ can adjust the merged model such that interference is minimized. In contrast, Task Arithmetic (TA), with SAR varying across tasks, exhibits interference that cannot be reduced below the norm of the orthogonal component.
> > >
> > > We also experimentally show that interference, measured as L1 distance between the final embeddings of task-specific and merged models (following [1]), for Iso-C is lower than the interference for TA for ViT-B/16: https://imgur.com/a/b9Lpk8q.
> > >
> > > [1] Representation Surgery for Multi-Task Model Merging, ICML 2024
> > >
> > > >*search for a larger $\lambda$, this is unstable on LLM.*
> > >
> > > It is true that the singular value average reduces the Frobenius norm and we need to search for a larger $\lambda$. However, we did not observe instabilities for $\alpha \in [0.5, 3.1]$ (plot: https://imgur.com/a/GgB6nFD) in NLP experiments on T5-Large -- a 770M parameter LLM (see response to Rev. mnUL, Sec. C3).
> > >
> > > >*checkpoints used by ISO and TSV are different*
> > >
> > > Thank you for pointing out this important detail. We use checkpoints introduced by Consensus Merging in all the experiments in our paper (both for our and competing methods) providing a fair comparison.
> > >
> > > However, many other papers use TA checkpoints, and we were not aware of this when comparing with additional methods during this rebuttal. We reran Iso-C and Iso-CTS using the TA checkpoints to fairly compare with methods that reported merging using them:
> > >
> > > ||ViT-B/32|ViT-L/14|
> > > |-|-|-|
> > > |Fisher|68.3|83.7|
> > > |RegMean|71.8|82.2|
> > > |PCB|76.3|87.5|
> > > |CART|83.0|90.8|
> > > |**Iso-C**|_84.1_|_92.5_|
> > > |**Iso-CTS**|**84.3**|**93.0**|
> > >
> > > Iso-C and Iso-CTS still outperform all of the added methods.
> > >
> > > >*compare with EMR merging or Twin merging.*
> > >
> > > We would like to highlight that, during this rebuttal, we added comparisons with 4 vision methods, including the recent SOTAs CART and PCB, as well as PEFT evaluations — e.g. recent KnOTS — and NLP experiments. Moreover, we consider merging methods that result in a **single set of multi-task weights and do not change the inference procedure**, which can be used as a drop-in replacement for the pre-trained model. Twin-Merging, however, composes task-specific components at test-time and alters the inference algorithm increasing its cost over two times. Similarly, EMR-Merging uses additional per-task parameter masks and rescalers to perform inference. We will include this discussion in the revised manuscript.

---

### Official Review · Reviewer_BR4L · 2025-03-12

**Overall Recommendation:** 4

**Summary:**

This paper proposes a novel model merging framework that enhances alignment between subspace of task models and merged model. The framwork includes two algorithms, (1) Iso-C that achives isotropic by flattenning the spectrum to the averaged singular values and (2) Iso-CTS in which lowest spectral components are further replaced by task-specific directions. Experiments on merging 8, 14, and 20 CLIP models demonstrated the effectiveness of this framework.

## update after rebuttal

Thank you for the very detailed and complete rebuttal. I appreciate authors' newly added theoretical justification and arguments under responses to Reviewer ff9S. Also, thank you for providing the comparisons to those new baselines. It clearly shows the advantages of Iso-C and Iso-CTS. The new results authors provided for LoRA FT models and also T5 models (under responses to Reviewer mnUL) are also convincing. The new results provided during the rebuttal have greatly improved the quality of the paper and I hope the authors can include these results in the final version. I have updated my scoring to reflect this.

**Claims And Evidence:**

Yes

**Essential References Not Discussed:**

N/A

**Experimental Designs Or Analyses:**

The multi-task model merging experimental designs and the metrics for evaluating merged model follow the standard procedure of model merging work. The three analyses experiments in Section 5.3 are sound and valid.

**Methods And Evaluation Criteria:**

Yes

**Other Comments Or Suggestions:**

- Iso-CTS requires multiple SVD operations, can authors provide a complexity analysis of the proposed method?
- Why Table 1 only reports “average absolute accuracy” and “average normalized accuracy” but not the proposed NAI?
- The authors should provide more motivation for the performance gains by Iso-C and Iso-CTS and include supporting numerical results to echo "no task left behind". e.g., in Figure 3(a), tasks that were less represented in the TA model (such as Cars, DTD, and SUN397) exhibit greater performance improvement after Iso-C, which aligns with expectations.

**Other Strengths And Weaknesses:**

### Strengths

- Both Iso-C and Iso-CTS exhibit low hyperparameter sensitivity, as demonstrated in the analysis and appendix.
- The proposed framework enhances model merging performance as well as fairness.
- The proposed framework uses subspace alignment ratio as the metric for quantifying subspace similarity.
- The paper is generally clear, and the overall structure is easy to follow.

### Weaknesses

- Lacks theoretical justification or motivation on why enhances the average subspace alignment ratio by making the merged matrix isotropic.
- The SVD baseline is insufficient.

[1] Choi, J., Kim, D., Lee, C., & Hong, S. (2024). Revisiting weight averaging for model merging. arXiv preprint arXiv:2412.12153.

- The main experiments are conducted on a single model family.
- Current results are based on merging fully fine-tuned models. Evaluating the approach on PEFT models (e.g. LoRA) would provide a more complete understanding of methods' capabilities.
- As acknowledged by the authors in the Limitations section, the methods have not been tested in the NLP domain.

**Questions For Authors:**

- The claim in Section 3.2 (lines 145–147) is somewhat confusing. TA suspected their effectiveness arises from the cosine similarity between the vectorized representations of the task matrices being close to zero. From Fig2(a), they are indeed close to zero?

**Relation To Broader Scientific Literature:**

Yes. This paper provides an analysis of how an isotropic merged matrix can enhance model merging performance, complementing existing spectral-based merging methods by showing solely modifying singular values can be a powerful approach. This work also analyze individual task performance, offering a fresh evaluation angle in areas such as fairness.

**Theoretical Claims:**

N/A

---

> ### Author Rebuttal · Authors · 2025-03-31
>
> We are glad that the Reviewer appreciates the soundness of the experimental protocol and analyses, the effectiveness of the proposed approach, and clear writing. We thank the Reviewer for their comments and we respond below.
>
> > [Weakness 1 (W1)]: *Lacks theoretical justification...*
>
> We provide a detailed explanation why making merged matrix isotropic increases subspace alignment in the response to Reviewer **ff9S** (section Q2).
>
> >[W2]: *SVD baseline is insufficient.*
>
> Thank you for pointing out CART which is a recent and very relevant baseline. We added it (alongside baselines suggested by Reviewer **ff9S**, section Q3) to our main Table for 8 tasks (as these are the only results reported in the CART paper):
>
> ||ViT-B/32|ViT-L/14|
> |-|-|-|
> |Fisher|68.3|83.7|
> |RegMean|71.8|82.2|
> |PCB|76.3|87.5|
> |CART|83.0|90.8|
> |**Iso-C**|**86.3**|_94.2_|
> |**Iso-CTS**|_86.2_|**94.7**|
>
> Iso-C and Iso-CTS outperform all these methods.
>
> >[W3]: *Experiments on a single model family.*
>
> In this rebuttal we add NLP experiments highlighting the effectiveness of Iso methods on the T5 -- an encoder-decoder language transformer (see the response to Reviewer **mnUL**, section C3).
>
> >[W4]: *...Evaluating the approach on PEFT (e.g. LoRA)...*
>
> Thank you for the suggestion - it helps to emphasize the generalizability of our approach. We follow the evaluation protocol of KnOTS[1], a recent SOTA (ICLR 2025) method for merging LoRA fine-tuned models, tested on 8 vision tasks using ViT-B/32 and ViT-L/14. For comparison, we merge the task-specific LoRA weights - provided by the authors - to the pre-trained models, and then we apply Iso-C and Iso-CTS. Below, we present the average normalized accuracy:
>
> ||ViT-B/32|ViT-L/14|
> |-|-|-|
> |KnOTS-TIES|68.0|78.2|
> |KnOTS-DARE|63.9|75.6|
> |**Iso-C**|_74.4_|_89.4_|
> |**Iso-CTS**|**75.0**|**89.6**|
>
> Iso-CTS achieves SOTA results in LoRA merging setting. Note that our method is a general purpose merging technique while KnOTS is specifically designed for the LoRA merging. This highlights the versatility of Iso methods.
>
> [1] Stoica et al. Model merging with SVD to tie the Knots, ICLR 2025.
>
> >[W5]: *The methods have not been tested in the NLP domain.*
>
> See our response to Reviewer **mnUL** (section C3) for NLP results.
>
> >[Comment 1 (C1)]: *...can authors provide a complexity analysis?*
>
> Let $\Delta_t \in \mathbb{R}^{m\times n}$, with $m\geq n$ and let $T$ and $L$ be the number of tasks and network layers, respectively. For simplicity, assume that each layer has a single matrix, whose dimensions are $m$ and $n$. In the analysis below, the lines refer to Algorithm 2 in the main paper.
>
> - Iso-CTS: One SVD on $\Delta_{TA}$ (lines 2-3) whose complexity is $O(mn^2)$ and this is applied to each layer, so the complexity is $O(Lmn^2)$; one SVD on each $\Delta_{t}, t= 1..T$, for each layer (line 5) so the complexity is $O(LTmn^2)$; finally line 11 requires two SVDs on matrices $U \in \mathbb{R}^{m \times r}$ and $V \in \mathbb{R}^{n \times r}$. Since $m \geq n$, then $r=n$, the complexity is $O(2Lmn^2)$. The total complexity is:
> $$O(IsoCTS)=O(Lmn^2+LTmn^2 + 2Lmn^2)=O(LTmn^2).$$
> - Iso-C: One SVD on $\Delta_{TA}$ with complexity:
> $$O(IsoC)=O(Lmn^2).$$
> - TSV (our nearest competitor): $T$ SVDs per layer on each task matrix (line 1 Alg. 1, TSV paper), and two additional SVDs per layer(line 10-11 Alg.1 TSV paper) and thus:
> $$O(TSV)=O(LTmn^2+2Lmn^2)=O(LTmn^2).$$
>
> While Iso-CTS and TSV share the same asymptotic complexity, Iso-CTS incurs slightly more overhead due to the SVD on $\Delta_{TA}$ (lines 2-3). Both methods can be further optimized by computing Truncated SVDs for Iso-CTS (line 7) and TSV (line 1 Alg. 1, TSV paper), since only a few components are retained. This reduces the complexity for both approaches. Iso-C is the most computational efficient algorithm.
>
> >[C2]: *Why Table 1 only reports “average absolute accuracy” and “average normalized accuracy” but not NAI?*
>
> We report these two metrics to stay consistent with previous literature (Consensus TA, TSV-M). In the revised manuscript we will add a Table reporting NAI.
>
> >[C3]: *The authors should provide more motivation...*
>
> Thank you for the suggestion. In the revised manuscript we will put more emphasis on "no task left behind" achieved by Iso methods highlighting higher performance improvements for tasks underrepresented in TA.
>
> >[Question 1 (Q1)]: *The claim in Section 3.2...*
>
> Yes, each pair of task vectors has a near-zero cosine similarity (Fig. 2a). However, our analysis goes a step further by comparing the cosine similarity between individual task vectors and the task arithmetic vector, demonstrating that this measure alone does not correlate with normalized accuracy improvement (Fig. 2b). Since cosine similarity alone does not explain performance gains, we introduce SAR. Unlike cosine similarity, SAR allows for meaningful differentiation among task matrices by highlighting shared subspaces (Fig. 3b). Additionally, SAR positively correlates with NAI (Fig. 3a).

---

### Official Review · Reviewer_ff9S · 2025-03-13

**Overall Recommendation:** 4

**Summary:**

This paper focuses on bridging the performance gap between the merged and task-specific models. They first show that the subspace alignment of merged and task-specific models correlates with performance improvement. Then, they propose an isotropic merging method to improve the merging performance via flattening the singular values. An extension is proposed to further improve the alignment and performance by considering the task-specific subspaces. Empirical results show that their method consistently outperforms the baselines.

## After rebuttal:

I think most concerns are well addressed after the rebuttal. I particularly like the interesting findings of the reasons that leads to performance gap in TA. I believe those insights are valuable and critical to the community. Theoretical analysis is provided to enhance the understanding. Experimental design and results are also improved.

I have no questions now. Just suggest that the authors summarize all the reviews and appropriately integrate them in the paper, no matter in the main paper or the appendix. I am very glad to raise my score to 4.

**Claims And Evidence:**

I am confused about Fig.2, where the authors propose their motivation. The motivation itself makes sense to me, but Fig.2 is confusing. See questions for details.

---

**After rebuttal:**

I think the claims are clear to me right now and are supported by sufficient evidence.

**Essential References Not Discussed:**

N/A

**Experimental Designs Or Analyses:**

There is a dataset in which the performance is inconsistent. Also, the improvement of Iso-CTS over Iso-C is not sufficiently discussed. See questions for some comments.

---

**After rebuttal:**

Concerns are well addressed.

**Methods And Evaluation Criteria:**

I think some important baselines are missing. The motivation of the method is unclear to me as well. See questions for details.

**Other Comments Or Suggestions:**

N/A

**Other Strengths And Weaknesses:**

The overall organization and writing of this paper are good. I also like the findings that the subspace alignment is critical to model merging, which I believe is important to the understanding of model merging.  My main concern is two-fold. First, it lacks some important baselines. Second, there are some potential issues regarding the motivations, the analysis of experimental results, and the effectiveness of the extended method. Please see the questions for details.

---

**After rebuttal:**

I think most concerns are well addressed after the rebuttal. I particularly like the interesting findings of the reasons that leads to performance gap in TA. I believe those insights are valuable and critical to the community. Theoretical analysis is provided to enhance the understanding. Experimental design and results are also improved.

**Questions For Authors:**

1. In Fig.2, the author tries to show that the task vector orthogonality is not the reason for TA performance improvement. However, Fig.2(b) is confusing to me. The x-axis is the similarity between $\Delta_i$ and $\Delta_{TA}$. Why do we need to compare these similarities? Fig.2(a) shows the task-vector orthogonality, but 2(b) shows some bad NAI, which is enough to conclude the incorrelation.
2. I may be missing or misunderstanding something, but I have a question regarding the motivation of the proposed method. In Sec.3.3, the author shows that the merging performance is correlated with $SAR_{avg}$. However, in Sec.4.1, the author uses Fig.1(a) as their motivation to “flatten” the singular values. I wonder what the relationship is between these two motivations. The authors claim that the variability of $SAR_{avg}$ is due to the skewness in Fig.1(a), but I didn't see a clear relationship.
3. Some important baselines are missing in the experiments, such as Fisher merging, RegMean, and a recent SOTA PCB-merging. Recent literature also competes with them. Is there any reason the authors do not compare their method with those baselines?
4. What are the values of $k$ in Tab.1?
5. In Fig.4(c), the performance of SUN397 is worst with $\beta=0.5$. Though this is the only inconsistency, I am curious if there is any explanation.
6. In Fig.5(a), the performance of Iso-C improves significantly when $\beta \to 1$, while in Fig.6, the improvement of Iso-CTS is marginal compared to Iso-C. Does it mean that the most improvement is due to the isotropic singular values but not the design in Alg.2? I am curious about the performance of Iso-CTS w/o line 12 (i.e., when there are no isotropic singular values).

---

I have no questions now. Just suggest that the authors summarize all the reviews and appropriately integrate them in the paper, no matter in the main paper or the appendix. I am very glad to raise my score to 4.

**Relation To Broader Scientific Literature:**

Model merging is a popular method to construct a multi-task model without retraining. However, the performance gap between the task-specific and merge models is critical. This paper first analyzes the key reason why the merged model has a worse performance and then proposes a novel method to address it. I believe this paper can help enhance the understanding of model merging and be applied as an effective method to merge models.

**Theoretical Claims:**

N/A

---

**After rebuttal:**

They propose some theoretical analysis, which I believe will strengthen the interpretability of this paper and provide insightful views.

---

> ### Author Rebuttal · Authors · 2025-03-31
>
> We are pleased that the Reviewer appreciates the novelty of the proposed method, the significance of our contribution, and the clear writing. We thank the Reviewer for their constructive feedback, and below we respond to specific points raised.
>
> > [Question 1 (Q1)]: *In Fig.2, the author tries to show...*
>
> Providing Fig. 2(a) along with only the Normalized Accuracy Improvement (NAI) for each task would show that all task vectors $\Delta_t$ exhibit similar cosine similarities close to zero, yet the NAIs vary significantly (e.g. the DTD task vector is orthogonal to all others but has the lowest NAI). We agree that this alone suggests that mere orthogonality among task vectors does not explain differences in performance.
>
> Fig. 2(b) however takes this analysis one step further by examining whether the cosine similarity between each $\Delta_t$ and the task arithmetic vector $\Delta_{TA}$ correlates with NAI. Intuitively, one might expect that a higher similarity between $\Delta_t$ and $\Delta_{TA}$ would result in a higher NAI, but no clear correlation is found. This reinforces the observation that cosine similarity is not a good predictor of performance improvement. We conducted this analysis because we believe that a key factor in understanding the effectiveness of task arithmetic model merging is to directly compare each task matrix with the *merged* model matrix. Since cosine similarity between task vectors and task arithmetic vector alone does not explain performance gains, we propose analyzing the Subspace Alignment Ratio (SAR) between each individual and merged task matrices, which indeed shows a positive correlation with performance improvement (Fig. 3). In the revised manuscript we will include elements of this discussion to more clearly link the results in Figs. 2 and 3.
>
> > [Q2]: *I may be missing or misunderstanding something, but I have a question regarding the motivation of the proposed method...*
>
> The subspace alignment ratio $SAR_\text{avg}(\Delta_t, \Delta_{TA})$ quantifies how well a task matrix $\Delta_t$ is represented by the subspace of the task arithmetic matrix $\Delta_{TA}$. The subspace dimension, $k_M$ as defined in Eq. (6), is determined by the number of singular vectors required to minimize the reconstruction error in terms of Frobenius Norm. Because the singular value spectrum of $\Delta_{TA}$ is skewed, only a few singular values are large, leading to a low $k_M$ (see Fig. 4(a) - $\beta=0.0$ (TA), marked by a vertical red dashed line). Relying on these few singular vectors to represent each task matrix produces a highly variable $SAR_{avg}$ across tasks (Figure 4(b) - $\beta=0.0$), indicating that some tasks are not well captured by this limited subspace.
>
> The motivation for "flattening" the singular values via Iso-C in Sec. 4.1 is to address this issue. By scaling the singular values, the influence of less dominant singular values increases while that of the dominant ones decreases. This adjustment raises the effective subspace dimensionality $k_M$ (as shown by the vertical dashed lines for $\beta > 0$ in Figure 4a), resulting in a subspace that better represents all task matrices. Consequently, this leads to a higher $SAR_{avg}$ (Fig. 4(b)) and Normalized Accuracy Improvement (Fig. 4(c)). Thus, the skewness in Fig 1(a) explains the variability in $SAR_{avg}$ (and hence merging performance), and the singular value flattening is introduced as a solution to this limitation.
>
> > [Q3]: *Some important baselines are missing in the experiments, such as Fisher merging, RegMean, and a recent SOTA PCB-merging...*
>
> Thank you for pointing out PCB method which is a recent and relevant baseline. Originally, we omitted Fisher Merging and RegMean for brevity as they are outperformed by many recent methods. We additionally include the CART baseline requested by Reviewer **BR4L**. The PCB paper reports the average absolute accuracy for merging 8 tasks across 2 model sizes and we compare these results in the Table below:
>
> |Method|ViT-B/32|ViT-L/14|
> |---|---|---|
> |Fisher|68.3|82.2|
> |RegMean|71.8|83.7|
> |PCB|76.3|87.5|
> |CART|83.0|90.8|
> |**Iso-C**|**86.3**|_94.2_|
> |**Iso-CTS**|_86.2_|**94.7**|
>
> Iso-C and Iso-CTS outperform all of the added baselines.
>
> > [Q4]: *What are the values of $k$ in Tab.1?*
>
> We use $\frac{k}{r} = 0.8$ as a default for all the experiments (see L408-410, right column), where $r$ is a number of singular values for a given layer. Therefore, $k$ can vary across layers according to the $r$ for each particular layer.
>
> > [Q6]: *...I am curious about the performance of Iso-CTS w/o line 12...*
>
> We present the comparison of the performance of Iso-CTS and Iso-CTS w/o line 12: https://imgur.com/a/39CrGKJ. We observe that isotropic scaling is indeed a crucial component of Iso-CTS. However, the design in Alg. 2 also plays an important role, especially when the number of merged models increases, leading to up to 2.8% improvement on 20 tasks (see Table 1).

---

> > ### Comment · Reviewer_ff9S · 2025-04-03
> >
> > Thanks a lot for your detailed response! While most of my concerns are addressed, I still have a question regarding your response to Q2.
> > Based on the response, my understanding is that a skewed spectrum leads to a lower $k_M$, which further results in a higher $SAR$. However, the latter relationship is still unclear to me.
> >
> > From your response, the logic of "causal chain" is $\lbrace\sigma_i\rbrace_i \to k_M \to SAR$. But I don't know why a lower $k_M$ leads to a higher $SAR$ from Eq.5 and 6. The results in Fig.4 that you mentioned only imply that $\lbrace\sigma_i\rbrace_i \to SAR$ **or** $k_M \to SAR$, but I am still unsure whether it is due to a lower $k_M$.
> >
> > While I think this does not influence the quality and contribution of this paper, I'd still like to know whether $k_M$ affects $SAR$ and how it does so. Could you explain a bit (theoretically) based on Eq.5 and 6? Empirical results are also acceptable but I think it could be hard to verify it via experiments. I am glad to raise my score if this can be addressed.
> >
> > Due to the limited number of communication rounds, I'd like to summarize my review here. I really like the findings of this paper, esp. Fig.2 and 3, which provide new views to understand task arithmetic. Though some inclarities may be due to my misunderstanding, I strongly encourage the authors to make it clearer for readers, as other reviewers also post similar questions. Overall, this is an interesting and solid paper.

---

> > > ### Author Response · Authors · 2025-04-07
> > >
> > > We thank the Reviewer engaging with us in this discussion. Below, we formally clarify the causal chain $\{\sigma_i\}\rightarrow k_M\rightarrow SAR$.
> > >
> > > The Subspace Alignement Ratio (SAR) between a task matrix $\Delta_t$ and a merged matrix $\Delta_M$ is:
> > >
> > > $$\text{SAR}(\Delta_t,\Delta_M;k_M)=\frac{||\Pi_{k_M,M}\Delta_t||_F}{||\Delta_t||_F},$$
> > >
> > > where $\Pi_{k_{\text{M}},\text{M}}=U_{k_{\text{M}},\text{M}}U^\top_{k_{\text{M}},\text{M}}$ is the projection onto the subspace spanned by the top $k_{\text{M}}$ left-singular vectors of $\Delta_{\text{M}}$. The rank $k_M$ minimizes the approximation error:
> > >
> > > $$k_{M}=\text{min}\lbrace k:||\Delta_M-\Pi_{k,M}\Delta_M||_F\leq\epsilon||\Delta_M||_F\rbrace.$$
> > >
> > > ### **$\sigma_i\rightarrow k_M$: The connection between the skewness of the spectrum of $\Delta_{M}$ and $k_M$**
> > >
> > > Using the SVD, $\Delta_{M}=U\Sigma V^T$, where $\Sigma=\text{diag}(\sigma_1,\ldots,\sigma_r)$, by definition of Frobenius norm we have:
> > >
> > > $$\Vert \Delta_M\Vert_F^2=\sum_{i=1}^r\sigma_i^2,\quad\Vert\Delta_M-\Pi_{k,\text{M}}\Delta_M\Vert_F^2=\sum_{i=k+1}^r\sigma_i^2.$$
> > >
> > > Hence, the relative approximation error becomes:
> > > $$\frac{\Vert\Delta_M-\Pi_{k,M}\Delta_M\Vert_F^2}{\Vert\Delta_M\Vert_F^2}=\frac{\sum_{i=k+1}^r\sigma_i^2}{\sum_{i=1}^r\sigma_i^2},$$
> > >
> > > and $k_M$ can be defined as:
> > >
> > > $$k_M=\text{min}\left\lbrace k:\frac{\sum_{i=k+1}^r\sigma_i^2}{\sum_{i=1}^r\sigma_i^2}\leq\epsilon^2\right\rbrace.$$
> > >
> > > This formulation is equivalent to the one used in the paper but **explicitly shows how the skewness of the spectrum $\lbrace \sigma_i\rbrace$ controls $k_M$**. When $\Delta_M$ has a skewed spectrum (e.g. $\sigma_1^2 \gg \sum_{i=2}^r \sigma_i^2$), a small $k_M$ is enough to satisfy the condition. This explains why Task Arithmetic $\Delta_{TA}$ ($\beta=0$ in Fig. 4(a)) — which has a skewed spectrum — yields a smaller $k_M$ than Iso-C, whose flatter spectrum leads to a larger $k_M$. We believe that expressing $k_M$ directly in terms of singular values highlights the link between the spectral skeweness  and subspace dimensionality. We will adopt this definition in the revised version.
> > >
> > > ### **$k_M \rightarrow SAR$: The connection between $k_M$ and SAR**.
> > >
> > > The rank $k_M$ defines the **effective rank** of the subspace identified by the merged model, determined directly by its spectrum. Let $k_{TA}$ be the effective rank of $\Delta_{TA}$, and define
> > >
> > > $$T=\lbrace u_1,..,u_{k_{TA}}\rbrace$$
> > >
> > > as the orthonormal basis formed by those $k_{TA}$ singular vectors. Flattening the spectrum of $\Delta_{TA}$ (Fig. 4(a)), yields $\Delta_{Iso-C}$ with  effective rank $k_{Iso}>k_{TA}$ (as discussed previously). This flattening modifies only the singular values of TA, leaving the singular vectors unchanged. Therefore, the original subspace $T$ is contained within the larger subspace spanned by the top singular vectors of $\Delta_{Iso-C}$, defined as:
> > >
> > > $$I=\lbrace u_1,..,u_{k_{TA}},..,u_{k_{Iso}}\rbrace.$$
> > >
> > > Thus, by construction, we have $T\subset I$.
> > >
> > > For simplicity, let $\Pi_T=\Pi_{k_{\text{TA}},\text{TA}}$ and $\Pi_I=\Pi_{k_{Iso},\text{Iso}}$ denote the projection operators onto the subspaces spanned by $T$ and $I$, respectively. Since $T\subset I$, for any matrix $\Delta_t$ it holds that:
> > >
> > > $$SAR(\Delta_t,\Delta_{TA})=\frac{\Vert\Pi_T\Delta_t\Vert_F}{\Vert\Delta_t\Vert_F}\leq\frac{\Vert\Pi_I\Delta_t\Vert_F}{\Vert\Delta_t\Vert_F}=SAR(\Delta_t,\Delta_{Iso-C}),$$
> > >
> > > This inequality holds because by definition:
> > > $$\frac{\Vert\Pi_T\Delta_t\Vert_F^2}{\Vert\Delta_t\Vert_F^2}=\frac{\sum_{i=1}^{k_{TA}}\sum_j\langle u_i,\Delta_t^{(j)}\rangle^2}{\Vert\Delta_t\Vert_F^2}\leq\frac{\sum_{i=1}^{k_{TA}}\sum_j\langle u_i, \Delta_t^{(j)}\rangle^2+\sum_{i=k_{TA}+1}^{k_{Iso}}\sum_j\langle u_i,\Delta_t^{(j)}\rangle^2}{\Vert\Delta_t\Vert_F^2}=\frac{\Vert\Pi_I\Delta_t\Vert^2_F}{\Vert\Delta_t\Vert^2_F},$$
> > >
> > > where $\Delta_t^{(j)}$ denotes the $j$-th column of $\Delta_t$. Equality holds (i.e. $SAR(\Delta_t,\Delta_{TA}) = SAR(\Delta_t,\Delta_{Iso-C})$ ) only if the additional vectors added to the basis $T$ — that is $\lbrace u_{k_{TA}+1},\ldots,u_{k_{Iso-C}}\rbrace$ — are orthogonal to each $\Delta^{(j)}_t$ or, equivalently, if they do not intersect the column space of $\Delta_t$ (i.e. its left sigular vectors).
> > >
> > > Hence, in general **a lower $k_M$ yields smaller or equal SAR than a larger $k_M$**. However, our empirical findings show that enriching the basis $T$ with singular vectors corresponding to smaller singular values in original task arithmetic spectrum (i.e. $\lbrace u_{k_{TA}+1},\ldots,u_{k_{Iso-C}}\rbrace$) **consistently increases the alignement ratio** (Fig. 4(b)), implying that these vectors are relevant for representing each task matrix $\Delta_t$ and not orthogonal to its left singular vectors. This analysis formally supports the claim that a higher effective rank $k_M$ for the merged matrix leads to a higher SAR. We will make explicit the connection between $k_M$, Iso-C and SAR at the end of Section 4.1. in the final version of the paper.

---

### Official Review · Reviewer_mnUL · 2025-03-14

**Overall Recommendation:** 4

**Summary:**

The paper studies how to improve model merging methods by leveraging the singular value decomposition (SVD) of task matrices, defined as the differences between fine-tuned models' weight matrices and the pre-trained model. The authors first show that merging performance correlates with the alignment between the top eigenspace of task-specific and merged matrices. Building on this insight, they propose *isotropic merging* (ISO-C), which replaces the singular values of merged matrices with a uniform spectrum.  Additionally, they refine the merged matrices by substituting directions associated with small singular values with task-specific eigenspaces orthogonal to the top eigenspace of the merged matrices before flattening the spectrum. These approaches achieve state-of-the-art performance on standard computer vision model merging benchmarks.

## Update after rebuttal

I maintain my positive assessment.

**Claims And Evidence:**

The claims are well supported.

**Essential References Not Discussed:**

N/A

**Experimental Designs Or Analyses:**

The experimental design is sound.

**Methods And Evaluation Criteria:**

The paper includes 3 standard model merging benchmarks with 8, 14, and 20 tasks and evaluates 3 CLIP models with VIT base and large encoders, as it is standard in this research area.

**Other Comments Or Suggestions:**

- Have you analyzed SAR at different depths of the models?
- I assume Fig. 3(a) is obtained for a ViT base. How would it look for the larger model?
- Finally, I think the paper would be much stronger presenting some results for NLP as well, as it is standard in the field since model merging is widely relevant beyond computer vision.

**Other Strengths And Weaknesses:**

Other strengths:
- The proposed method is original and contributes novel insights into model merging.
- Model merging via weight interpolation is a relatively recent but impactful area. Improving it by reducing task interference is a significant contribution.
- The paper is clearly written and well-structured.

Minor weaknesses:
- Sec. 3, along with Fig. 2 and 3, lacks details of the experimental setting and the models considered.
- $\rho$ (L195, right) is undefined in the main text.
- L240-246 repeatedly mention *more/less correlated tasks*. I think this terminology is vague and should be clarified in terms of alignment.
- The claim in L256-258 (right) would benefit from explicit justification.

**Questions For Authors:**

N/A

**Relation To Broader Scientific Literature:**

The paper contributes to the ongoing research on improving model merging and mitigating task interference. Similarly to concurrent studies, it considers SVD decomposition of the weight matrices. The proposed techniques – uniform singular value scaling and selective incorporation of task-specific subspaces – are novel and improve performance over previous techniques across model sizes and the number of merged tasks.

**Theoretical Claims:**

N/A

---

> ### Author Rebuttal · Authors · 2025-03-31
>
> We are pleased that the Reviewer acknowledges the novelty of the proposed method, the contribution towards understanding model merging and clear writing. We thank the Reviewer for the constructive feedback, and below we respond to specific points raised.
>
> > [Minor weakness 1 (MW1)]: *Sec. 3, along with Fig. 2 and 3, lacks details of the experimental setting and the models considered.*
>
> We used ViT-B/16 and 8 tasks (see Sec. 5.1 for details). We will clarify this in the revised manuscript.
>
> > [MW2]: *$\rho$ (L195, right) is undefined in the main text.*
>
> $\rho$ is a Pearson correlation coefficient – defined in the caption of Fig. 3. We will unify the notation to $\rho_{\text{TA}}$ and add the definition to the main text.
>
> > [MW3]: *L240-246 repeatedly mention more/less correlated tasks. I think this terminology is vague and should be clarified in terms of alignment.*
>
> We will clarify this part in the revised manuscript to avoid vagueness:
>
> *However, significant variability in the average alignment ratio across the dataset leads to a lower accuracy improvement for **less aligned tasks** compared to **the tasks belonging to groups of high alignment**. This variability stems from the skewness of the task arithmetic spectrum (Fig. 1), which is concentrated in the first few singular values (which we call top or dominant), favoring **the tasks from the highly aligned groups**.*
>
> > [MW4]: *The claim in L256-258 (right) would benefit from explicit justification.*
>
> We can formalize the SVD problem for the first left principal singular vector as the variance maximization problem:
>
> $u_1=\arg\max_{||u|\vert=1}||\Delta_{TA}^Tu||^2=u^T\left(\sum_{t=1}^T\Delta_t\Delta_t^T\right)u+u^T(\sum_{{t,s=1,t\neq s}}^T\Delta_t\Delta_s^T)u$
>
> If a particular task $\Delta_j$ has dominant directions with significantly lower intensity compared to the other tasks (i.e. lower Frobenius Norm), then its individual contributions $\Delta_j \Delta_j^T$ to the total variance becomes smaller. Similarly, cross terms involving $\Delta_j$ will also be comparatively small.  Therefore, task $j$ explicitly contributes less to the maximized variance captured by the first principal singular direction.
>
> Moreover, if the directions of $\Delta_j$ are orthogonal or nearly orthogonal to $u_1$, (i.e. $u_1^T\Delta_j=0$), task $j$ contributes minimally or not at all along this principal direction. Similar considerations apply to subsequent singular vectors $u_2, \ldots u_k$, defining the common subspace. Finally, as the number of tasks $T$ increases and tasks become more diverse, it becomes increasingly likely that tasks with distinct but smaller-magnitude directions will be underrepresented or absent in the dominant singular directions identified by the task arithmetic decomposition. This is empirically supported by our results. Iso-CTS provides the most improvement when the number of tasks increases.
>
> > [Comment 1 (C1)]: *Have you analyzed SAR at different depths of the models?*
>
> We analyze SAR at different depths. For the ViT-B/16 model, we calculate SAR between fine-tuned and merged weight matrices and an average of all the matrices from a given layer. We present the results here: https://imgur.com/a/tLnEoAi. We observe that the alignment is higher for Iso across all layers of the vision transformer. One may expect early layers to be more aligned but we find that for both approaches the alignment is similar across the layers.
>
> > [C2]: *I assume Fig. 3(a) is obtained for a ViT base. How would it look for the larger model?*
>
> Yes, we obtain Fig. 3(a) for ViT-B/16. See the Figure for ViT-L/14 here: https://imgur.com/a/3V6xv7T. It closely resembles Fig. 3(a) for ViT-B/16 from the paper.
>
> > [C3]: *I think the paper would be much stronger presenting some results for NLP...*
>
> We present NLP results following the experimental setup from [1]. We use T5-Large-LM-Adapt base model fine-tuned on tasks from T0 mixture. We consider subsets of 8 and 7 NLP tasks adhering to the setup from Table 1 from [1] and compute an average accuracy of Iso-C and Iso-CTS in these settings:
>
> |**Method**|**8 tasks (Zhou et al., 2022)**|**7 tasks (Yadav et al., 2023)**|
> |-|-|-|
> |Weight Avg.|56.4|60.5|
> |TA|63.8|69.2|
> |TIES|62.8|71.9|
> |Fisher|57.7|61.0|
> |RegMean|69.1|74.3|
> |MaTS|72.5|81.5|
> |**Iso-C**|**75.6**|**83.3**|
> |**Iso-CTS**|_75.2_|_82.8_|
>
> Both Iso-C and Iso-CTS significantly outperform the competing approaches, which highlights the versatility of our proposed methods. We observe that Iso-CTS achieves slightly worse results than Iso-C. This is consistent with our vision results in which both approaches performed very similarly when merging 8 models. We argue that the common space captures all the directions necessary to reliably represent these 7 and 8 NLP tasks, while task-specific subspaces may became more effective when merging more models.
>
> [1] Tam et al. Merging by Matching Models in Task Parameter Subspaces, TMLR 2024

---

> > ### Comment · Reviewer_mnUL · 2025-04-08
> >
> > Thank you for the response. I maintain my positive assessment.

---

### Decision · Program_Chairs · 2025-05-01

**Decision:**

Accept (poster)

**Comment:**

The work proposes to improve model merging by reducing task interference. The method computes the SVD of the summed task vectors (which are actually per-layer weight matrices) and then flattens the singular values to their average.

Strong improvements on standard computer vision model merging datasets are shown, yielding a similar performance to the recent Task Singular Vectors (TSV) method.

Reviewers unanimously liked the paper, but several issues were pointed out:
- Reviewer ff9S, Reviewer BR4L, Reviewer 38Gq found the motivation why flattening the singular values works unclear.
- Reviewer mnUL, Reviewer BR4L pointed out the lack of comparisons on NLP datasets.
- Reviewer BR4L pointed out a missing complexity analysis and lack of comparison to another SVD-based baseline.

The authors addressed all these points in their rebuttal, and all reviewers raised their score to 4. I recommend acceptance of the work.